# Unifying the mechanism of mitotic exit control in a spatiotemporal logical model

**Rowan S. M. Howell**[1,2], **Cinzia Klemm**[3], **Peter H. Thorpe**[3]*, **Attila Csikász-Nagy**[2,4]*

**1** The Francis Crick Institute, London, United Kingdom, **2** Randall Centre for Cell and Molecular Biophysics, King's College London, London, United Kingdom, **3** School of Biological and Chemical Sciences, Queen Mary University, London, United Kingdom, **4** Faculty of Information Technology and Bionics, Pázmány Péter Catholic University, Budapest, Hungary

* p.thorpe@qmul.ac.uk (PHT); attila.csikasz-nagy@kcl.ac.uk (ACN)

## Abstract

The transition from mitosis into the first gap phase of the cell cycle in budding yeast is controlled by the Mitotic Exit Network (MEN). The network interprets spatiotemporal cues about the progression of mitosis and ensures that release of Cdc14 phosphatase occurs only after completion of key mitotic events. The MEN has been studied intensively; however, a unified understanding of how localisation and protein activity function together as a system is lacking. In this paper, we present a compartmental, logical model of the MEN that is capable of representing spatial aspects of regulation in parallel to control of enzymatic activity. We show that our model is capable of correctly predicting the phenotype of the majority of mutants we tested, including mutants that cause proteins to mislocalise. We use a continuous time implementation of the model to demonstrate that Cdc14 Early Anaphase Release (FEAR) ensures robust timing of anaphase, and we verify our findings in living cells. Furthermore, we show that our model can represent measured cell–cell variation in Spindle Position Checkpoint (SPoC) mutants. This work suggests a general approach to incorporate spatial effects into logical models. We anticipate that the model itself will be an important resource to experimental researchers, providing a rigorous platform to test hypotheses about regulation of mitotic exit.

## Introduction

The ordering of mitotic events is tightly controlled in eukaryotes in order to ensure accurate chromosome segregation and prevent aneuploidy, a hallmark of cancer [1]. The Mitotic Exit Network (MEN) is a signalling network in *Saccharomyces cerevisiae* that interprets spatial and temporal signals in late mitosis, ensuring mitotic exit and cytokinesis occur only after proper segregation of the genetic material (reviewed in [2,3,4,5,6,7]). Since the network was first described by Jaspersen and colleagues [8], over 100 papers have been published on the topic. This volume of research has driven the MEN to become one of the best understood signalling pathways; however, it also poses a challenge to synthesise this knowledge. In this article, we propose a compartmental, logical model of the MEN that aims to represent a unified view of the network and make predictions about its behaviour.

**Data Availability Statement:** All relevant data are within the paper and its Supporting Information files.

**Funding:** The authors acknowledge funding from the Francis Crick Institute, which receives its core

funding from Cancer Research UK (FC001003), the UK Medical Research Council (FC001003), the Wellcome Trust (FC001003). The funders had no role in study design, data collection and analysis, decision to publish, or preparation of the manuscript.

**Competing interests:** The authors have declared that no competing interests exist.

**Abbreviations:** APC, Anaphase-Promoting Complex; ASC, Anaphase Specific Component; CDK, Cyclin-Dependent Kinase; DIC, differential interference contrast; FEAR, Cdc14 Early Anaphase Release; GBP, GFP-Binding Protein; GEF, Guanine nucleotide Exchange Factor; MEN, Mitotic Exit Network; NLS, Nuclear Localisation Signal; ODE, Ordinary Differential Equation; PDF, Probability Distribution Function; PKN, Prior Knowledge Network; SAC, Spindle Assembly Checkpoint; SC, synthetic complete; SIN, Septation Initiation Network; SPB, Spindle Pole Body; SPoC, Spindle Position Checkpoint.

Progression of the cell cycle in eukaryotes is controlled by the activity of Cyclin-Dependent Kinase (CDK). CDK activity begins low in G1 phase before increasing as cells enter S-phase and reaches its peak in mitosis [3]. In order for the cell to complete the cell cycle and return to its G1 state, it must reduce CDK activity and reverse phosphorylation of its substrates. In yeast, unlike other model eukaryotes, this occurs in a 2-step process [5]. Firstly, some cyclins, including Clb5 and to a lesser extent Clb2 [9], are destroyed at the metaphase–anaphase transition through the activity of the Anaphase-Promoting Complex (APC) in its Cdc20-coupled isoform. Then at exit from mitosis, the phosphatase Cdc14 is released from the nucleolus and dephosphorylates multiple CDK phosphosites around the cell, in particular Cdh1, an alternative APC subunit capable of targeting mitotic cyclins for destruction. This leads to complete reversal of mitotic CDK phosphorylation, permitting division of mother and daughter cells, and returns the cell to a G1 state.

This 2-step process of CDK destruction in mitosis is required in budding yeast as its morphology necessitates 2 checkpoints rather than the one normally found in eukaryotic cells. The Spindle Assembly Checkpoint (SAC) is found in most eukaryotes and ensures the kinetochores are correctly attached to chromosomes before spindle elongation occurs and acts through control of the APC subunit Cdc20. In contrast, the Spindle Position Checkpoint (SPoC) is not present in most eukaryotes. The SPoC acts to ensure both mother and daughter cells receive the full complement of genetic material after cytokinesis, by delaying mitotic exit until alignment of the spindle with the mother–bud axis.

The 2-step process of CDK inactivation is mirrored by the regulation of Cdc14 localisation. Throughout the majority of the cell cycle, Cdc14 remains tightly sequestered in the nucleolus, through its interaction with Net1. In early anaphase, Cdc14 is released into the nucleus by a group of proteins known as the Cdc14 Early Anaphase Release (FEAR) network (Fig 1). This release is not sufficient to initiate exit from mitosis, and in fact is transitory as Cdc14 will return to the nucleolus if the SPoC remains in place [10]. MEN activity, unlike the FEAR network, leads to sustained release of Cdc14, which, in turn, activates the alternative APC subunit, Cdh1, and the CDK inhibitor, Sic1, leading to cytokinesis and the entry of the cell into a new reproductive cycle [3]. The MEN is controlled by the SPoC and indirectly by the SAC, as the high level of CDK activity in metaphase prevents MEN activation.

Control of localisation is key to the regulation of the MEN. MEN-activating factors, such as Lte1, reside in the bud, while MEN-inhibiting proteins like Kin4 are restricted to the mother compartment [11,12]. Alignment of the spindle with the mother–bud axis is signalled by the entry of a Spindle Pole Body (SPB) into the bud [6,13]. MEN proteins localise to the SPB and so sense the change from the mother to the bud compartment, leading to MEN activation. The MEN pathway culminates in activation of the Mob1-Dbf2 kinase complex (Fig 1). The exact mechanism by which Mob1-Dbf2 promotes mitotic exit is not yet fully understood; however, it is known that the complex enters the nucleus [14], phosphorylates Net1 [15], and phosphorylates Cdc14 near to its NLS (Nuclear Localisation Signal), allowing Cdc14 to leave the nucleus [16].

The volume of research into mitotic exit makes it difficult to provide a unifying view of the process using only informal models, and so we turn to formal mathematical models which can account for this complexity. There are a number of published mathematical models of mitotic exit in budding yeast, but none represent the full extended network with spatial detail. Some of these are comprehensive Ordinary Differential Equation (ODE) models of the cell cycle, which include some details of the MEN, for example, Chen and colleagues [17] and Kraikivski and colleagues [18]. While these models represent some aspects of mitotic exit control, their broad scope means they lack detail. The Queralt model [19,20] was the first mathematical model built to focus on the regulation of Cdc14. This work was used as a basis for further modelling,

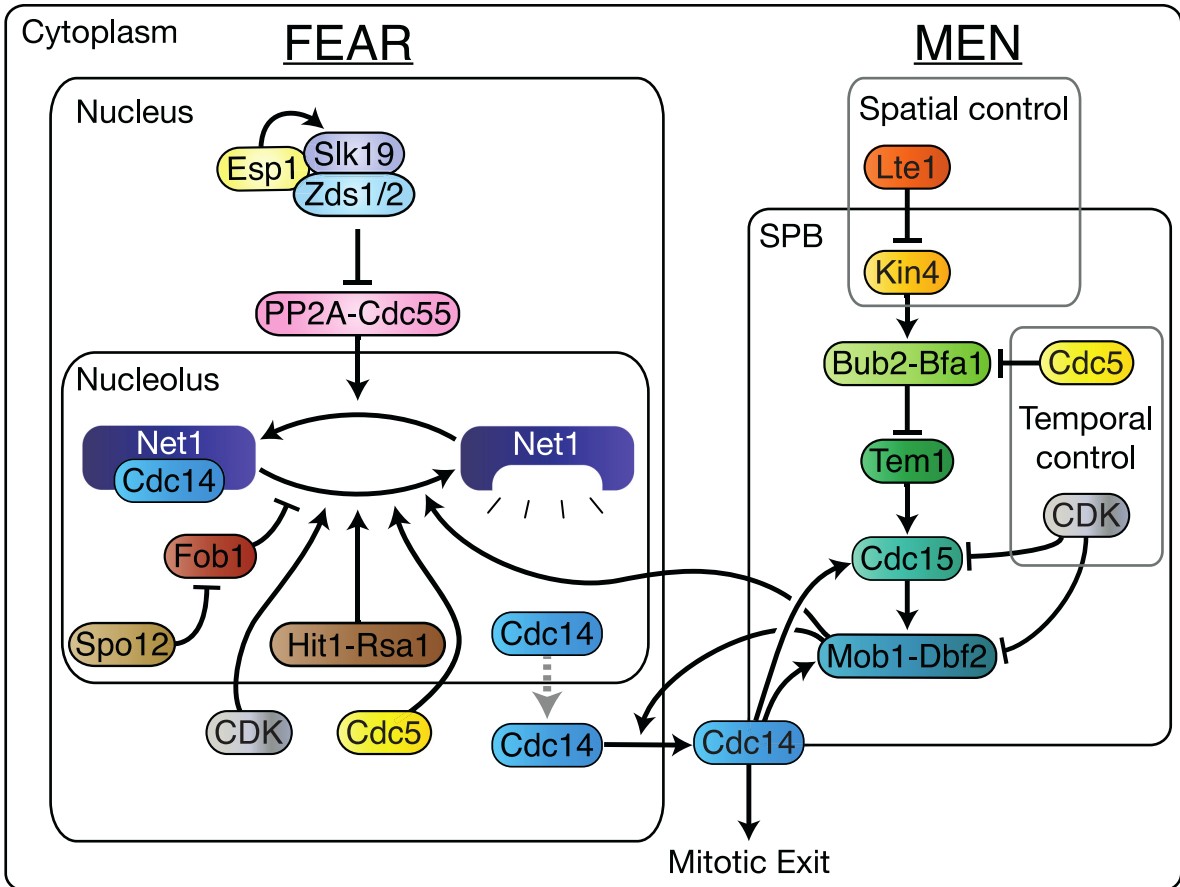

**Fig 1. A simplified graphical description of the MEN and FEAR pathways.** A detailed description of both networks can be found in S1 Text. CDK, Cyclin-Dependent Kinase; FEAR, Cdc14 Early Anaphase Release; MEN, Mitotic Exit Network; SPB, Spindle Pole Body.

with Vinod and colleagues [21] and Hancioglu and Tyson [22] expanding the Queralt model to examine the interplay between the MEN and the FEAR network. These ODE models were important to develop the biological understanding of the MEN; however, they are unable to represent spatial regulation. A compartmental ODE model of the SPoC was used by Caydasi and colleagues [23] to explore the spatial aspects of Tem1 and Bub2-Bfa1 regulation; however, its scope was limited to a small number of proteins. We aim to build on these models to create a model of the extended network in spatial detail.

A major limitation of ODE models is their scalability; a large ODE model requires many parameters which must be experimentally measured or inferred, which is virtually impossible for larger networks. Logical models are an alternative modelling formalism which attempts to avoid the issue of parameterisation by focusing on network structure, producing qualitative, binary predictions about activity of proteins (reviewed in [24,25]). Generalised logical models are an extension of the Boolean formalism; the nodes of a logical model may take any number of discrete states, instead of the 2 (0 or 1) permitted in Boolean models. Many tools for the construction and simulation of logical models have been developed, with an emphasis on interoperability (CoLoMoTo.org). Logical models have been used widely to explore cell cycle regulation in yeast, starting with Li and colleagues [26], and some large scale models have included coarse-grained representations of exit from mitosis [27,28]. A logical model was used to explore regulation of the Septation Initiation Network (SIN) [29], the MEN's homologous

pathway in *Schizosaccharomyces pombe* [30]. Münzner and colleagues [31] recently published the most comprehensive logical yeast cell cycle model yet. However, this model does not explicitly represent spatial effects in the network, and 3 out of 12 genetic phenotypes relating to core MEN proteins that the authors tested were found not to match the experimentally determined phenotypes. Some variations of the logical modelling formalism have included spatial effects, for example, the EpiLog tool [32], which models cell–cell signalling. However, there is currently no logical formalism capable of representing intracellular spatial regulation. To overcome these limitations, we have created a novel compartmental logical modelling framework that can represent spatial effects.

We have used this compartmental logical modelling framework to build a model of mitotic exit control. Our proposed model aims to represent the above aspects of regulation of Cdc14 localisation from metaphase to telophase. In recent years, further roles for the MEN have been proposed in the control of spindle alignment [33] and in cytokinesis [34]; however, we consider these processes to be beyond the scope of the model. Furthermore, we model only the process of initiation of mitotic exit and not its execution, so phenomena such as Cdc14 endo-cycles that rely on degradation of Cdc5 during mitotic exit [35,36] are outside the scope of the model. The model provides an account of changes in the localisation and activity of key regulators of mitotic exit from metaphase to late anaphase. In particular, it can predict the conditions leading to release of Cdc14 and mitotic exit and how these vary in mutant strains.

## Materials and methods

### Logical modelling

In the logical modelling formalism, proteins are nodes in a network, with edges between nodes representing their regulation (activation or inhibition) [25]. Therefore, a logical model is a directed, signed network along with a set of logical (ANDs and ORs) rules describing how the different kinds of regulation interact with each other (Fig 2B). Any logical rule may be written in disjunctive normal form, as a single OR over multiple ANDs. This simplifies interpretation of the rule, and so as far as possible, we have used this representation. The final ingredient required for a logical model is an update scheme, describing when nodes have their state updated. The simplest update scheme is a synchronous scheme where all nodes are updated at the same time. Attractors are sets of network states which, once entered, cannot be left under the synchronous update scheme. The simplest attractors are steady states, which are states that once entered, the model will never leave. There are also higher-order attractors containing multiple states; a model in such a state will cycle through multiple intermediate states in a given order. It can be shown that all logical networks have at least 1 attractor. While the synchronous update scheme is simple, it is unrealistic, as in real biochemical systems, some reaction will always occur first [25]. Therefore, asynchronous update schemes, in which nodes are updated independently in a predefined or random order, give more realistic dynamics. Any steady state under the synchronous update scheme will remain steady under an asynchronous update scheme, although this is not always true of higher-order attractors.

### Compartmental logical modelling

In a compartmental ODE model, the concentration of each protein in each compartment is described by separate variables. Similarly, in a compartmental logical model, a "localization" node exists for each protein in each compartment it is permitted to localise to. The state of this node corresponds to whether the protein is present in this compartment (1) or not (0). In addition to these localisation nodes, an activity node for each protein and each compartment exists to track whether the protein is active in this compartment (1) or not (0). Activation of the

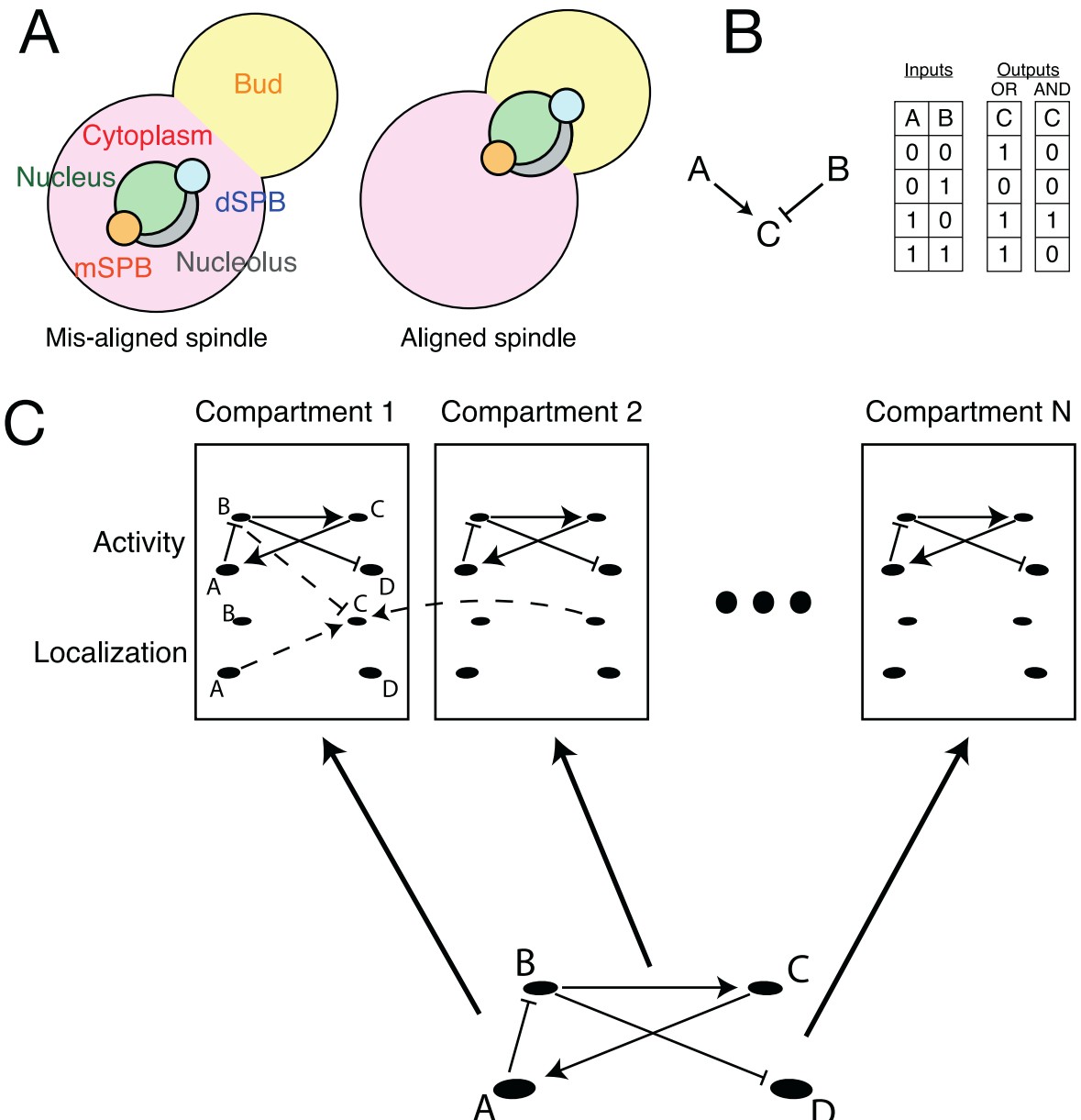

**Fig 2. The compartmental logical modelling framework and the MEN.** (A) The compartments present in the model. (B) Illustration showing how a simple biochemical motif can be interpreted as a set of logical rules, shown as a truth table. (C) Schematic showing how a logical network can be expanded across multiple compartments, with additional rules to describe the regulation of localisation. dSPB, daughter SPB; MEN, Mitotic Exit Network; mSPB, mother SPB.

localisation node is a prerequisite for activation of the activity node. The rules of the network are then built from an activity network—describing how proteins control each other's enzymatic activity through posttranslational modifications—and a localisation network—describing how proteins control each other's localisation to different compartments. The resulting network can be expressed as a logical network, albeit one where each protein appears multiple times for each compartment. This means that compartmental logical networks are larger than the underlying networks. If there are $n$ proteins represented in the model and $C$ compartments, the resulting compartmental logical network has $2 \times n \times C$ nodes.

## Model construction

The FEAR network was trained using the CellNOptR package [37], run 100 times in parallel, with max inputs per gate set to 4. Activity and localisation networks as well as a set of location specific rules were created in BoolNet format, and custom R scripts were used to generate the compartmental logical model. Briefly, the relevant nodes were created for each of the permitted locations, and then the rules were read in and distributed across the nodes by creating an edge list. This edge list included modifications, such as requiring the localisation node to be on for the activity node to be switched on. Generally, protein activity can only be regulated by proteins in the same compartment. The exceptions are the SPB and nucleolus, proteins in these compartments can be regulated either by proteins here or in the cytoplasm or nucleus, respectively. The "Spindle alignment" node controls whether proteins at the dSPB (daughter SPB) can be regulated or exchange with the cytoplasm ("Spindle alignment" = 0) or with the bud compartment ("Spindle alignment" = 1). All R scripts may be accessed at https://github.com/RowanHowell/CLM-R.

## Simulation of mutants

To analyse the phenotypes of mutant strains, we placed all mutations into one of the following categories:

1. **Hyperactive**, an allele which is resistant to inhibition and will be active wherever it is localised.

2. **OE**, overexpression either by the *GAL1* promoter or provision of the gene on a multicopy plasmid, the protein is present and active everywhere. This high level of expression of a protein can break the usual rules of protein regulation (see below).

3. **KD**, (knock-down), a functionally inactive allele, often temperature or analogue sensitive, which is inactive wherever it is localised.

4. **Delete** or **Deplete**, either deletion of the protein or depletion via a conditional promoter, localisation prevented everywhere.

5. **Location** or **! Location**, the forced localisation of the protein in 1 compartment or (!) the prevention of that localisation.

6. **Phosphomutants**, phosphomutants represent a rewiring of the network and therefore were considered on a case-by-case basis.

Of the above, most are straightforward; however, overexpression is more complex. In some cases, the sheer quantity of the overexpressed protein can alter the wiring of the network, for example, activating a downstream component despite the presence of an inhibitor [38]. To account for these effects, we introduced additional overexpression nodes for each protein. In the case where a protein activates its downstream components in a way that could be blocked by an inhibitor, the overexpression node circumvents this inhibition (S1B Fig). However, note that localisation of a protein in a compartment is always a necessary requirement for that protein's activity in that compartment. Similarly, overexpression of an inhibitor may block the activation of a protein even in the presence of an activator. The only exceptions to this are the cases of nonphysiological conditions of high Tem1 activity and lack of low level of Bub2 and Bfa1 activity in models 3 and higher. These modifications were applied to the edge list described above and were used to produce a variant of the model, usually suffixed "OE." Similarly, we treated forced localisation at the SPB as equivalent to a local overexpression, so a

model variant suffixed "SPB" included the forced localisation nodes needed to model this pertubation.

## Discrete time dynamics and steady states

We represented the logical model as a Boolean network using the Van Ham mapping [39], in which each level of activity is represented as an individual node, in order to make use of computational tools designed for Boolean networks. As synchronous steady states are necessarily steady states in an asynchronous setting, all steady states were identified in the synchronous system. This was performed by solving the satisfiability problem using the BoolNet package for R [40], which employs the PicoSAT solver [41], based on the algorithm of Dubrova and Telsenko [42]. In some cases, multiple steady states exist for the same cell cycle stage; in order to determine the phenotype in such situations, we used Monte Carlo simulations of the asynchronous model with nodes chosen uniformly at random, again using BoolNet functions. Unless otherwise stated, all simulations were performed from the same physiological initial conditions (S1 File) and were ran until either steady state was reached, the "Mitotic Exit" node was activated, or the number of time steps reached 10,000. All R scripts may be accessed at https://github.com/RowanHowell/CLM-R.

## Continuous time dynamics

We used the MaBoSS package [43], using scripts developed for the MaBoSS python package [44], to perform continuous time simulations of the logical model. The BoolNet model was converted to MaBoSS format using the GINsim tool [45]. Rate parameters were fit to match experimentally determined length of mitosis in wild type, *bub2Δ*, and *kin4Δ* cells (S6 Fig and S1 Table, [46]). Spindle alignment times were simulated as a Brownian motion, using a custom Python script. All Python scripts may be accessed at https://github.com/RowanHowell/CLM-Python.

## Model accessibility

Model 5 can be accessed in SBML and BoolNet format at https://github.com/RowanHowell/CLM-R and Models 5 and 6 can be accessed in MaBoSS format at https://github.com/RowanHowell/CLM-Python. Model 5 has been uploaded to the BioModels database [47] (ID: MODEL2007200001).

## Yeast strains and methods

Yeast was cultured in standard growth media with 2% (w/v) glucose at 30°C, unless otherwise stated. All yeast strains are derivatives of BY4741, unless otherwise specified. Plasmids were constructed by gap repair either through in vivo recombination or the NEBuilder plasmid assembly tool (New England Biolabs, United States of America). Linear products were created by PCR with primers from Sigma Life Science and Q5 Polymerase (New England Biolabs). The strains and plasmids used in Fig 5D were a gift from Gislene Pereira [48], with the exception of the empty plasmid, which was pWJ1468. mRuby2-TUB1 strains were constructed using a linearised plasmid, "pHIS3p:mRuby2-Tub1+3′UTR::URA3, which was a gift from Wei-Lih Lee (Addgene plasmid # 50639; http://n2t.net/addgene:50639; RRID: Addgene_50639) [49]. *NUD1-GFP* strains are derived from a library derived from BY4741 (*his3Δ1 leu2Δ0 met15Δ0 ura3Δ0*) [50,51]. *TEM1-YFP* and *CDC15-YFP* strains were constructed by homologous recombination from E438. *GBP* and *CLB2-CDC28-GBP* plasmids express their product from the *MET3* promoter and were derived from pWJ1512 [52]. All spot

assays show 10-fold serial dilutions, with cultures of the same optical density. Counterselection of plasmids with uracil selection was achieved by addition of 750-$\mu$g/ml 5FOA (FluoroChem, United Kingdom). Plasmids used in this study are listed in S2 Table; strains are listed in S3 Table.

## Fluorescence microscopy

In the anaphase length assay, cells were grown shaking overnight in synthetic complete (SC) media at 30°C, then diluted 1 in 10 in fresh SC media and left to grow for 3 hours. Cells were then transferred to a -uracil agarose cube and placed into a sealed chamber. *SPO12* and *spo12Δ* cells were placed in side-by-side chambers. The cells were preincubated at 30°C for an hour before imaging. Cells were imaged using a DeltaVision Elite (GE Healthcare, USA), with a 60× 1.42NA Oil Plan APO and an InsightSSI 7 Colour Combined Unit illumination system (CFP = 438 nm, mRuby2 = 575 nm). Images were captured with a front illuminated sCMOS camera, 2,560 × 2,160 pixels, 6.5-$\mu$m pixels, binned 2 × 2. Time-lapse videos were captured over 2 hours, with images captured at 2-minute intervals. Images were analysed using FIJI [53], with the Bio-formats plugin [54].

For the forced CDK localisation experiments, cells were grown shaking overnight in -leucine media supplemented with additional methionine at 23°C. They were then transferred to -leucine -methionine media and grown shaking for 4 hours at 23°C and then imaged. For the *NUD1-GBP* forced localisation experiments, functionality of the SAC was assessed as described in Fraschini and colleagues [55]. Cells were grown shaking overnight in 2% raffinose -leucine media. They were then transferred to 2% raffinose -leucine media for 2 hours before being spun down and washed in water. They were resuspended in 2% galactose YP media containing 3-$\mu$g/ml alpha factor. They were transferred to incubate shaking for 2 hours before the alpha factor (The Francis Crick Institute Peptide Chemistry STP) was washed out, and the cells were washed and resupended in 2% galactose YP media containing 15-$\mu$g/ml nocodazole (Sigma-Aldrich, Germany) and incubated shaking for 3 hours. In both assays, cell was imaged with a Zeiss Axioimager Z2 microscope (Carl Zeiss AG, Germany), with a 63× 1.4NA oil immersion lens and using a Zeiss Colibri LED illumination system (RFP = 590 nm, GFP = 470 nm). Bright field images were obtained and visualised using differential interference contrast (DIC) prisms. Images were captured using a Hamamatsu Flash 4 Lte. CMOS camera containing a FL-400 sensor with 6.5-$\mu$m pixels, binned 2 × 2. Images were analysed with ICY [56].

## ODE simulations

Simulations of the ODE model of Caydasi and colleagues [23] (BioModels database ID: BIOMD0000000702) were performed with Copasi [57], using the CoRC package. Parameters were unchanged from the original model, except initial conditions which were chosen to match the steady states of the prealignment model. Forced localisation of Bfa1 at the SPB was modelled by decreasing the off-rate of Bfa1 species by a factor of 1,000.

## Results

### Model construction

Due to the complexities of the spatial aspects of the model, we combined an expertise-based approach with a model-fitting approach to construct the model. The FEAR network, which acts only in the nucleus, was trained against a dataset of 50 mutant phenotypes using the Cell-NOptR tool [37]. The rest of the model was built from the literature, and the trained FEAR network was integrated into it.

CellNOptR uses a genetic algorithm to train a Boolean model against known phenotypes [58,37]. The genetic algorithm takes a Prior Knowledge Network (PKN) and evolves this network, using its fit to data as a measure of fitness, to optimise the model with respect to the dataset. We built a PKN comprising of 22 edges (S2 File) and trained it against a dataset of 52 mutants from 11 publications (S3 File), using the FEAR from the nucleolus as the output. A detailed, referenced description of the FEAR network can be found in S1 Text. We found that, due to the stochasticity of the genetic algorithm, there was significant variation between the fit achieved by independent runs of the algorithm. For this reason, we ran the algorithm 100 times, distributed in parallel and considered the optimal fits achieved. We found that several phenotypes, in particular those relating to Cdc5 overexpression, were difficult for the algorithm to fit. Cdc5, when expressed at a high level, is clearly capable of releasing Cdc14 (see, for example, [59]); however, the activity of Cdc5 is thought to be stable throughout late mitosis. Therefore, we reasoned that overexpression is likely to break the normal logic of Net1 inactivation. For this reason, we allowed overexpression of Cdc5 to "feed-forward" and regulate Net1 according to different rules than those for physiological levels of Cdc5. Practically, this meant we introduced a node called "Cdc5OE" which had the same outputs as Cdc5 in the PKN, this node was then treated as all the others in the training process. We found this allowed for the identification of models which could fit 88% of the training dataset. The "Cdc5OE" node was removed during integration with the MEN model, but its effect was recovered by the addition of overexpression nodes for each protein in the model. After performing training, we became aware of the Hit1-Rsa1 complex and its role in FEAR [60] and added Hit1-Rsa1 to the model.

A compartmental model is the product of an activity and localisation network superimposed (Fig 2C). We could not enforce the necessary conditions to use CellNOptR to train this kind of model, although in principle, an evolutionary algorithm could be used for this purpose. Instead, we decided to build the model by hand; a list of update rules for each node in the final model (Model 5) with details of the evidence for each can be found in S4 File, and a graphical representation is given in S1A Fig. A detailed, referenced description of the MEN can be found in S1 Text. The model has 6 compartments: the nucleus, nucleolus, cytoplasm (mother compartment), bud, mSPB, and dSPB (Fig 2A). In budding yeast, the old and new SPBs have some minor physiological differences, and it is the old SPB that enters the bud [61]. However, it has been shown that reversing this pattern has no significant effect on MEN signalling [62], and therefore, we consider the dSPB and mSPB to refer only to the destination of the SPB and not to their age-related identity.

A key decision was how to model the activities of CDK and Cdc14. CDK activity depends on the concentration of cyclins in the cell. Early mitotic and S-phase cyclins such as Clb5 are largely degraded by the APC in its Cdc20 isoform at the metaphase–anaphase transition. Consequently, the level of CDK activity in the cell decreases; however, late mitotic cyclins, such as Clb2, remain present until activation of the alternative APC subunit, Cdh1, at mitotic exit. In the interests of simplicity, we do not distinguish specific cyclin contributions; instead, the model has a high and low level of activity for CDK, representing the metaphase and anaphase levels of CDK, respectively. Similarly, the Cdc14 nodes are 2-levelled, as although FEAR release of Cdc14 is largely limited to the nucleus, its impact on MEN proteins means that a low level of Cdc14 must reach the cytoplasm in early anaphase. We also used 2 levels for Cdc15 nodes. There is limited evidence that the activity of Cdc15 changes throughout mitosis; instead, Cdc15 is thought to be controlled by its localisation, with geometric constraints and enzymatic funnelling allowing Cdc15 to phosphorylate Dbf2 only at the SPB. However, Cdc15 and CDK are thought to engage in a negative feedback loop, where Cdc15 phosphorylation of Nud1 allows CDK localisation at the SPB, which itself prevents Cdc15 localisation, closing the loop [63]. The localisation of active Tem1 at the SPB then allows a high level of Cdc15 to localise to

the SPB, leading to recruitment of Mob1 and activation of Dbf2. For reasons not yet understood, CDK is excluded from the SPB in the presence of active MEN components. We speculatively represented this effect by inclusion of a rule preventing CDK localisation at the SPB in the presence of active Mob1 and Dbf2. Therefore, the model includes a low level of Cdc15 capable of engaging in the Cdc15-Nud1-CDK feedback loop and a high level that is recruited by Tem1 and can activate Mob1-Dbf2. We modelled the activity of Kin4 as preventing Bfa1 localisation at the SPB; this is a simplification of the real mechanism in which Kin4 phosphorylation increases the turnover of Bfa1 at the SPB [64]. Both of these mechanisms have the same effect of keeping Bfa1 away from its inhibitor Cdc5 at the SPB.

We combined the FEAR and MEN models into a single compartmental logical model (Model 0, Table 1), treating the FEAR network as an activity network, with the exception of PP2A-Cdc55 [65] and Cdc14, which are controlled through localisation. The resulting network consists of 302 nodes (331 nodes in the model with overexpression). Although mitotic exit is executed through the concerted effort of many other proteins, such as Cdh1 and Sic1, the primary trigger for mitotic exit is release of Cdc14. The output of the model is therefore the full release of Cdc14, and we will treat this as synonymous with mitotic exit.

## Restriction of mitotic exit to anaphase

While developing the model, we tested it against a number of well-characterised mutants in order to refine its behaviour. In particular, we found that fitting phenotypes relating to the restriction of mitotic exit to anaphase posed a challenge for Model 0 and necessitated further development of the model. The MEN can be thought of as a coincidence detector, waiting for both temporal and spatial signals before allowing mitotic exit to occur [48,66,67]. However, there is still contention over how exactly MEN activity is restricted to anaphase. Two classes of mutants show MEN activity prior to anaphase: firstly, *bfa1Δ* or *bub2Δ* [68] and secondly, *CDC15-7A*, *MOB1-2A* [63,67,69]. These mutants differ in that deletion of *BUB2* or *BFA1* renders the MEN completely insensitive to spatial or temporal signals. On the other hand, the *CDC15-7A* mutation results in disruption of the SPoC [46] and when combined with the *MOB1-2A* mutation will exit mitosis at any point in anaphase, or upon spindle alignment in metaphase [67]. Initially, we tested our model against the phospho-null mutants by creating new networks in which edges joining CDK and Cdc14 to Cdc15 and Mob1 were removed.

**Table 1. Description of model versions.**

| Model | Modification |
|---|---|
| 0 | Base model. |
| 1 | As 0 and the low (anaphase) level of CDK inhibits Cdc15 loading in absence of Tem1. |
| 2 | As 1 and the ASC inhibits Cdc15 loading in metaphase, in the absence of Tem1 and CDK. |
| 3 | As 2 and multilevel Tem1, Bub2, and Bfa1. |
| 3a | As 3 and identification of ASC as Cdc5. |
| 4 | As 3 and Lte1 can inhibit Bfa1 activity in a mechanism parallel to Kin4. |
| 4a | As 3 and Lte1 can activate Tem1 activity and localisation. |
| 4b | As 3 and Lte1 can activate Tem1 activity but not localisation. |
| 5 | As 4 and identification of ASC as Cdc5. |
| 5a | As 5 and Cdc5 localisation at the SPBs depends on the high level of Bub2-Bfa1 localisation there. |
| 6 | As 5 but Lte1 regulation of Bfa1 can influence speed of Tem1 activation (MaBoSS implementation). |

ASC, Anaphase Specific Component; CDK, Cyclin-Dependent Kinase; SPB, Spindle Pole Body.

Our original model (Model 0) employs a simple rule for Cdc15 (high) localisation that depends only on Tem1 and not on other known regulators, such as CDK, Cdc14, or Cdc5. This simple model could not fit the behaviour of the *CDC15-7A* mutation (Fig 3). This was because this model relied on localisation of Tem1 at the SPB to allow full recruitment of Cdc15 there, meaning that disruption of the CDK regulation of Cdc15 would not change the spatial signalling. To address this, we created a new model (Model 1), in which Cdc15 could localise to the SPB in the absence of CDK. This model could now represent the behaviour of *CDC15-7A* cells but not the *CDC15-7A MOB1-2A* double mutant (Fig 3). Model 1 predicts that the double mutant should exit mitosis at any point during metaphase, whereas the correct behaviour in metaphase is to wait until spindle alignment [67]. This points to an additional level of regulation of Cdc15. We propose that Cdc15 can load onto SPBs in the absence of CDK or Tem1, in a way that is dependent on an Anaphase Specific Component (ASC). We introduced the ASC into a further model (Model 2) and found that this model is now capable of correctly representing the behaviour of the single and double phospho-mutants (Fig 3).

Having established that the model requires both CDK inhibition of Cdc15 localisation at the SPB and an additional level of regulation from an unknown ASC, we decided to test whether the model could predict the phenotype of *bfa1Δ*. We found that Model 2 predicted that *bfa1Δ* cells would exit mitosis in anaphase regardless of spindle alignment but not in metaphase (Fig 3). This is because in this model, Cdc15 and Mob1 are still inhibited by CDK. In order for the disruption of the upstream MEN components like Bfa1 to affect Cdc15 and Mob1, it is necessary for this state to be transmissible. Therefore, we introduced 2 levels of activity for Bub2, Bfa1, and Tem1 (Model 3). In our model of a wild-type cell, Bub2 and Bfa1 vary from a high level of activity, in which Tem1 is fully inhibited, to a low level, where Tem1 is sufficiently active to recruit Cdc15. When modelling *bfa1Δ* or *bub2Δ* cells, Tem1 becomes hyperactivated, allowing it to localise to the SPB independently of Bub2 and Bfa1 and recruiting Cdc15 despite the presence of high levels of CDK (Fig 3). This is justified by the finding that a constitutively active mutant *TEM1-Q79L* localises symmetrically to both SPBs independently of Bfa1 [70]. It is an unfortunate side effect of this modelling choice that the model suggests the level of Tem1 at the SPBs is elevated in a *bfa1Δ* or *bub2Δ* strain, whereas fluorescent measurements suggest the opposite [23]. Furthermore, in order for the hyperactivated state of Tem1 to transmit to Mob1, Model 3 allows this high level of Tem1 activity to combine with Cdc15 to recruit Mob1. Although Cdc15 can function in the absence of Tem1 [66], Tem1 and Cdc15 are found in complex under physiological conditions [11,71], suggesting that Tem1 may be able to play a role in recruiting Mob1. It is crucial that CDK inhibits the localisation and not the activity of Mob1 in order for the hyperactive state of Tem1 to be able to overcome this inhibition in metaphase in a *bub2Δ* or *bfa1Δ* strain.

## The SPoC in the absence of Kin4

Although the *kin4Δ* and *bub2Δ* mutants both lead to loss of SPoC function, the extent of loss of function differs between the two. This can be seen both in the proportion of cells that exit mitosis with misaligned spindles and the importance of the FEAR pathway in these cells [46]. Intriguingly, a *kin4Δ spo12Δ* double mutant has a fully functional SPoC [46], implying that spatial information is transmitted to the MEN via other mechanisms than just Kin4 activity. We tested the SPoC activity of *kin4Δ*, *spo12Δ* and double mutant cells in the model and found that in Model 3, the double mutant behaved like *kin4Δ* (Fig 4). In Model 3, the downstream effector of the MEN-activating zone is Kin4, so in its absence, there is no way for spatial information on spindle alignment to be transmitted to the MEN. A likely candidate for an additional spatial regulator of the MEN is Lte1 itself, and deletion of Lte1 in a *kin4Δ spo12Δ*

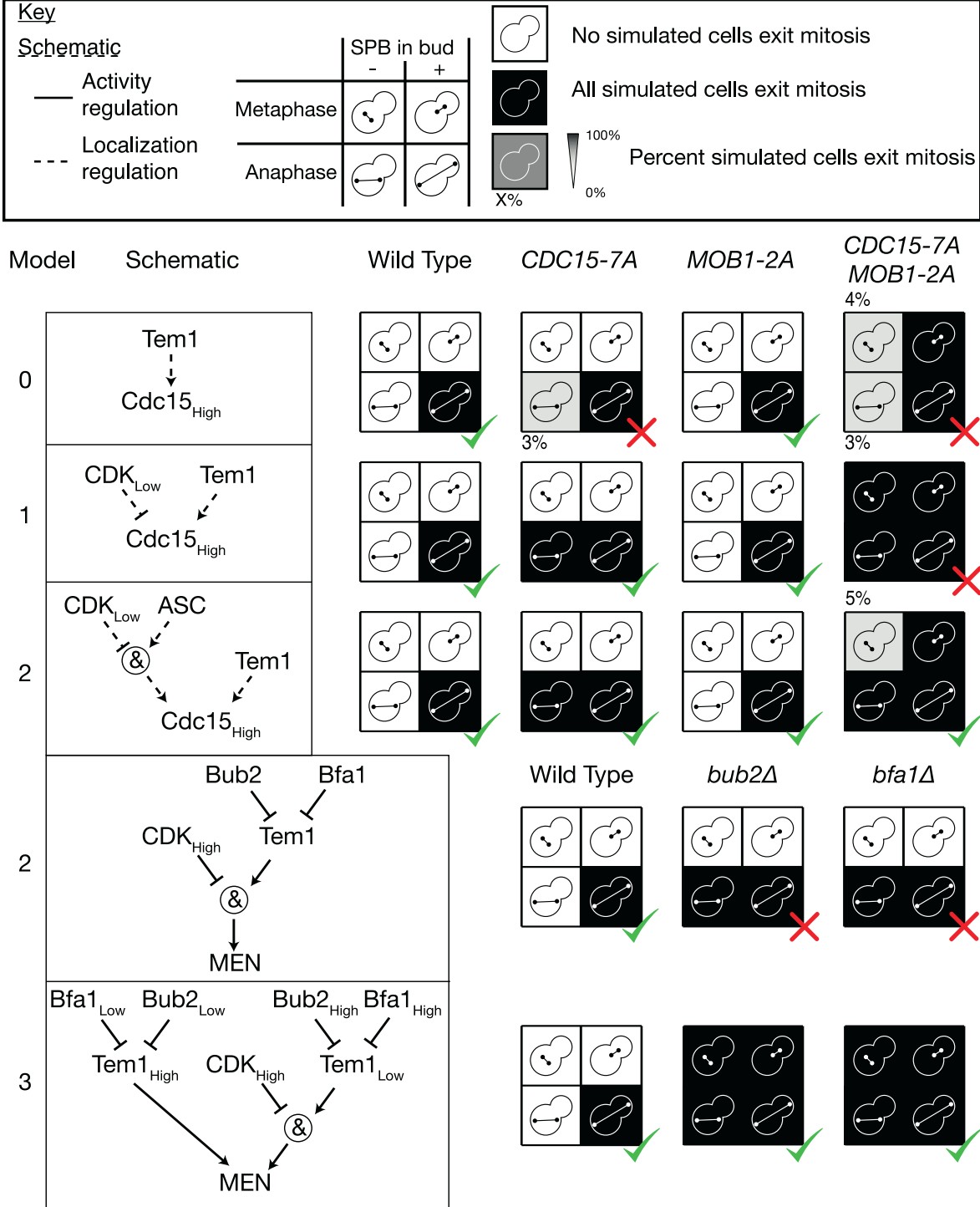

**Fig 3. Refinement of the MEN model based on mutants that can release Cdc14 in metaphase.** All simulations are performed using the random asynchronous update scheme; 100 cells were simulated for each mutant starting from realistic initial conditions. In the original model, the *CDC15-7A* mutant has an intact SPoC, in contradiction to the experimental evidence. Introducing regulation of the Cdc15$_{High}$ SPB localisation node by CDK fixes this issue (Model 1); however, this model cannot fit the behaviour of the *CDC15-7A MOB1-2A* double mutant. This double mutant can exit mitosis in metaphase but only when the spindle aligns and an SPB enters the bud [67]. The inclusion of an ASC that limits Cdc15 loading in metaphase resolves this problem (Model 2). Deletion of either component of the Bub2-Bfa1 GAP complex also permits exit from mitosis in metaphase; however, simulations of Model 2 do not agree with this. Introducing 2 levels of Bub2, Bfa1, and Tem1 activity (Model 3) is sufficient to represent this effect. All simulation data can be found in S5 File. ASC, Anaphase Specific

Component; CDK, Cyclin-Dependent Kinase; GAP, GTPase activating protein; MEN, Mitotic Exit Network; SPB, Spindle Pole Body; SPoC, Spindle Position Checkpoint. A green tick indicates that the simulated phenotype matches the phenotype from the literature, while a red cross indicates that the simulation deviates from the known phenotype.

background causes a significant delay in mitotic exit [46]. We created a new version, Model 4, in which Lte1 could inhibit Bfa1 activity in a way that depends on the phosphorylation of Bfa1. The introduction of this additional spatial regulation of the MEN meant that Model 4 could correctly represent both single and double *kin4Δ* and *spo12Δ* mutants (Fig 4). A version of the model (Model 4a) where Lte1 targets Tem1 activity and localisation directly could not correctly represent the phenotype of *KIN4* overexpression (S2 Fig). However, a version (Model 4b) where Lte1 targets Tem1 activity only is as effective at explaining these phenotypes as

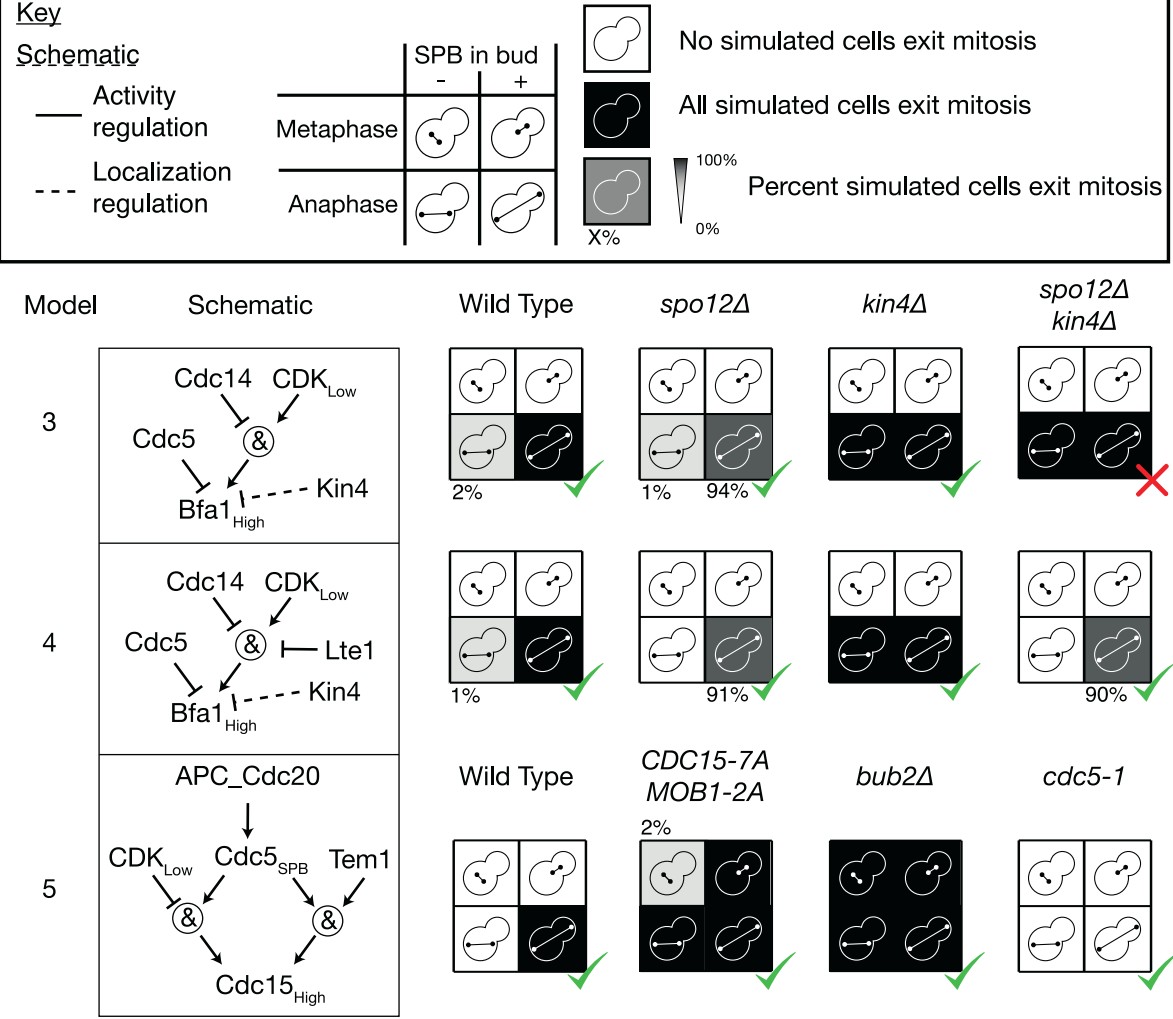

**Fig 4. Refinement of the MEN model based on the phenotype of *kin4Δ sp12Δ* cells.** All simulations are performed using the random asynchronous update scheme; 100 cells were simulated for each mutant starting from realistic initial conditions. In Model 3, the double mutant *kin4 spo12Δ* did not have a SPoC, in disagreement with experimental evidence [46]. By introducing an additional level of regulation of Bfa1 by Lte1, this issue was resolved in Model 4. This change also allowed for identification of the ASC with Cdc5, while maintaining the correct behaviour of related phenotypes, such as *CDC15-7A MOB1-2A, kin4Δ* and *cdc5-1*. All simulation data can be found in S5 File. APC, Anaphase-Promoting Complex; ASC, Anaphase Specific Component; CDK, Cyclin-Dependent Kinase; MEN, Mitotic Exit Network; SPB, Spindle Pole Body; SPoC, Spindle Position Checkpoint. A green tick indicates that the simulated phenotype matches the phenotype from the literature, while a red cross indicates that the simulation deviates from the known phenotype.

Model 4. Regardless of the specific rule used, it is clear that Lte1 acts through interruption of Bub2-Bfa1–mediated inhibition of Tem1.

## Cdc5 is the anaphase specific component (ASC)

We introduced the ASC to the model to allow for inhibition of Cdc15 localisation in metaphase. One candidate for this regulation is polo kinase Cdc5, which was identified as required for localisation of overexpressed Cdc15 at the SPB in the absence of Tem1 [66]. Fluorescently tagged Cdc5 is visible at the SPBs throughout mitosis; however, recent research suggests that it moves from the nuclear face of the SPB to the cytoplasmic face at the metaphase-to-anaphase transition [72,73]. Together, these findings suggest that Cdc5 could play the role of the ASC. When building Model 3, we considered Cdc5 as a candidate for the ASC (Model 3a) but found that before making the modifications in Model 4, this change prevented mitotic exit in *CDC15-7A MOB1-2A* cells with aligned spindles in metaphase (S3 Fig). However, the combination of identifying Cdc5 as the ASC and the regulation of Bfa1 by Lte1 from Model 4 resolves this issue (Model 5, Fig 4).

Botchkarev and colleagues [72] found that Cdc5 localisation at the SPB depends on Bub2-Bfa1. This is reflected in the rule for Cdc5 localisation at the SPB in Model 5, which depends on the low level of Bub2-Bfa1. We considered an alternative version (Model 5a) which depends on the high level of Bub2-Bfa1. This has the advantage of matching the asymmetric pattern of Cdc5 localisation observed by Botchkarev and colleagues [72]. However, it could no longer match the phenotype of *CDC15-7A MOB1-2A* cells (S3 Fig), as Cdc5 localisation at the SPB does not occur prior to spindle alignment in this model.

## Model validation

Having optimised the model's behaviour against specific mutants to create Model 5, we wanted to know whether the model could accurately predict the phenotype of the many mutants described in the literature. We found 147 mutants from 36 publications, representing overexpressions, knock-downs, deletions, and forced localisations of proteins present in the model. We classified these mutants depending on whether they exited mitosis in metaphase, anaphase pre-spindle alignment or anaphase post-spindle alignment. Mutants, which exited mitosis in anaphase pre-spindle alignment, were detected in a genetic background lacking either *KAR9* or *DYN1*, as these backgrounds result in a high frequency of cells with misaligned spindles. In some cases, we found disagreement between papers on particular mutants; in these cases, we chose a single finding to include in the dataset, prioritising experiments in the S288C/BY4741 genetic background, which were more numerous.

We simulated each of these mutants 100 times using the asynchronous update scheme and then calculated the percentage that exited mitosis, as judged by full release of Cdc14 into the cytoplasm. We then determined whether the majority of simulated cells exhibited the expected behaviour for each mutant in a given condition (S6 File). We found that the model correctly predicted the phenotypes of 81% (119/147) of mutants tested (Fig 5A).

Some of the phenotypes which the model predicted incorrectly relate to the rescue of temperature-sensitive alleles. For example, overexpression of Cdc5 can alleviate the temperature sensitivity of some MEN mutants, such as *tem1–3* [8], as well as cause release of Cdc14 during metaphase arrest [75]. However, while our model predicted that Cdc5 could cause unscheduled Cdc14 release, it required MEN proteins to do so and could not predict the alleviation of temperature sensitivity (Fig 5B). At an earlier point of testing, we found a similar effect with Spo12, another FEAR protein, which can rescue temperature sensitivity of MEN mutants, but our model predicted otherwise. A recent study showed that although Spo12 overpression can

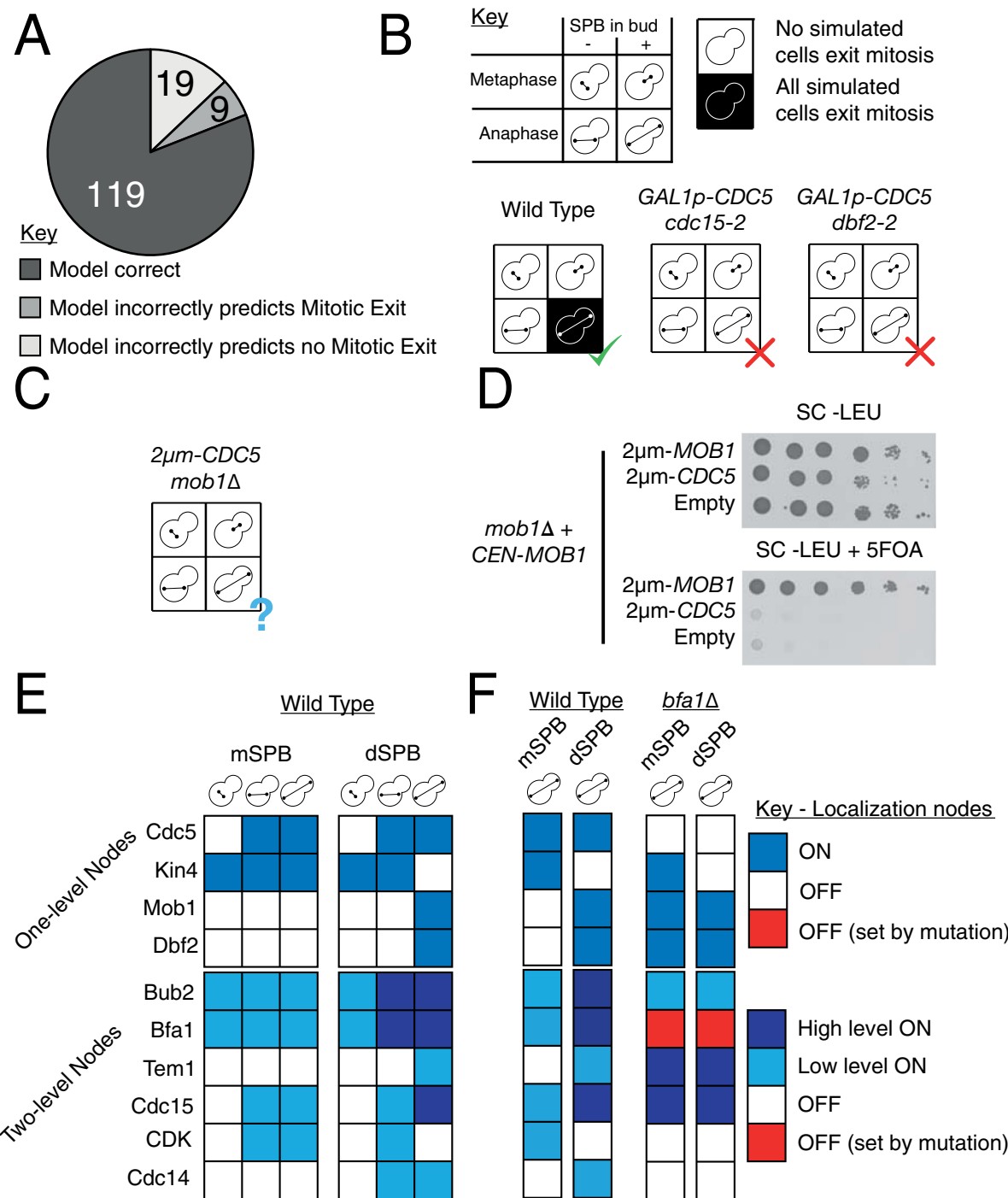

**Fig 5. Validation of Model 5 against literature phenotypes.** (A) The model correctly predicted **81%** of the 140 tested literature phenotypes (S6 File). (B) The model often failed at predicting the phenotype of cells with a genotype that mixes overexpression with other mutations, such as the rescue of the temperature-sensitive alleles *cdc15-2* and *dbf2-2* by overexpression of *CDC5*. (C) The model predicts that overexpression of *CDC5* cannot rescue the *mob1Δ* mutation. (D) Spot test confirming the model prediction that overexpression of *CDC5* cannot rescue full deletion of *MOB1*. A *mob1Δ* strain kept alive by provision of a *CEN-MOB1* plasmid with uracil selection was transformed with either a **2−μ**m plasmid bearing *MOB1* or *CDC5* or an empty plasmid. The *CEN-MOB1* plasmid was counterselected by addition of 5FOA, showing that moderate overexpression of *CDC5* is not sufficient for rescue of the *mob1Δ* phenotype. (E) The localisation state of MEN proteins on the SPBs in the 3 physiological stages of mitotic exit in the model. Steady states determined from synchronous update scheme. (F) Comparison of (a)symmetry of SPBs in the steady states of wild-type and *bfa1Δ* cells. All simulation data can be found in S5 File. CDK, Cyclin-Dependent Kinase; MEN, Mitotic Exit Network; SC-LEU, Synthetic Complete media lacking leucine.; SPB, Spindle Pole Body.

rescue temperature-sensitive MEN mutants, it cannot rescue deletion of these proteins [48]. Simulation of Spo12 overexpression combined with loss-of-function MEN mutations produced results that matched the phenotype of the full deletion of these MEN proteins, rather than the temperature-sensitive alleles. Altogether, this suggests that Spo12 overexpression allows mitotic exit to occur at a lower, but not zero, level of MEN activity. We suspected that a similar effect may explain the model's inability to predict the effects of Cdc5 overexpression. To test this, we transformed a *mob1Δ* strain, kept alive by provision of a plasmid encoding *MOB1*, with a 2-*μ*m plasmid expressing *CDC5*. When the *MOB1* plasmid was selected against using 5FOA, we found that the provision of *CDC5* from a 2-*μ*m plasmid could not suppress the lethality of *MOB1* deletion (Fig 5D), in agreement with the model's prediction (Fig 5C). This demonstrates that any effect caused by Cdc5 overexpression must rely on activation of at least the final part of the MEN pathway. More broadly, it suggests that suppression of partial or conditional lethality is a particular issue for our logical model, in which activity must be set to one of a number of discrete states.

While most phenotypes involving overexpression were correctly matched (40/60), many of the phenotypes which the model predicted incorrectly relate to overexpression (20/28), especially when combined with other mutations (19/28). We made certain decisions about how overexpression would interact with other aspects of regulation, for example, that an overexpressed protein could localise at the SPB without the proteins usually required for localisation there. This was inspired by the fact that deletion of Tem1, a protein required for Cdc15 localisation at the SPB, can be rescued by overexpression of *CDC15*, suggesting that high levels of Cdc15 can localise at the SPB independently of Tem1 [66]. However, there are counterexamples; overexpression of *KIN4* is lethal, as it prevents mitotic exit by inhibiting localisation of Bfa1 at the SPB, but this lethality can be rescued by deletion of the PP2A subunit Rts1, which controls localisation of Kin4 at the SPB [79]. There is no general rule which can account for both of these behaviours, without additional structural information relating to how these proteins bind to the SPB. This is likely to be a general problem when building compartmental logical models.

We found the attractors of the model fit closely with current understanding of the MEN. In the wild-type model, there is a single attractor for each of the stages of mitotic exit: metaphase, pre-spindle alignment, and post-spindle alignment. In these attractors, the patterning of proteins on the SPBs matches the known localisation patterns of MEN proteins (Fig 5E). Disruption of the asymmetrical distribution of MEN proteins, such as by the *bfa1Δ* mutation, is accurately captured by the model (Fig 5F).

Overall, the model fits the majority (>80%) of literature phenotypes, and the cases where it cannot represent the behaviour of real cells represent the limits of the logical modelling framework.

## Timing of mitosis

While mutations affecting the function of the FEAR network are not lethal, they reliably cause a delay to exit from mitosis [10]. We wanted to know whether our model could reflect this delay. In our model, Cdc15, Nud1, and CDK participate in a negative feedback loop, which is broken by the counteraction of CDK phosphorylation by Cdc14. Therefore, we predicted that loss of Cdc14 activity prior to full MEN activation would delay MEN activation.

Interpretation of timings in logical models can be difficult. With a synchronous update scheme, timings are meaningless, while an asynchronous update scheme has, at best, dimensionless pseudotime represented by the number of discrete time steps executed. However, more accurate than either of these is the translation of the model into a continuous time

Markov chain applied on the state space of the model [43], which can then be simulated with the Gillespie algorithm using the MaBoSS package [43,44]. In short, this means nodes are updated independently as a random process, occurring at a specified rate, meaning changes in the network state can be assigned continuous timings. Using MaBoSS, we simulated wild-type and *spo12Δ* cells in anaphase, post-spindle alignment (Fig 6A). For these simulations, we did not modify the standard rate parameters, as we wish only to compare the mutants to each other. We found that, as expected, the simulated FEAR mutant cells were significantly delayed in exit from mitosis. Intriguingly, the distribution of exit times in the FEAR mutant is not just shifted right, to longer times, but the shape of the distribution is also altered. The distribution of exit times for *spo12Δ* has a long tail (S5B Fig), indicating that in addition to the increased mean, the variance of the distribution is also increased.

We wanted to know whether this effect was detectable in real cells. Previous studies have used bulk measurements to demonstrate the delay caused by FEAR mutants [10]; however, to measure the distribution of exit times requires single cell measurements. We used a *CDC14-CFP mRuby2-TUB1* strain to quantify the length of mitosis in time-lapse videos. We defined entry into anaphase as the first frame showing an extended mitotic spindle and completion of mitosis by disassembly of the spindle (Fig 6B and 6C, S4 Fig, and S1 Video). We performed 5 time courses with side-by-side chambers containing *SPO12* and *spo12Δ* strains (S5A Fig). We found that there were some differences in the mean lengths of anaphase in the *SPO12* strain between time courses, possibly due to differences in the time cells spent in the chamber. Therefore, we normalised the times in each time course to the mean of the *SPO12* cells. Qualitatively, the experimental distributions of exit times quite closely matched the simulations (Fig 6D, S7 File), with the *spo12Δ* distribution shifted to the right with a heavier tail as compared to the *SPO12* distribution. To quantify this effect, we compared the Fano factor, a scale-free measure of variability, of the distributions (Fig 6E). We found that deletion of the FEAR component caused an increase in the Fano factor in both simulation and experiment. This suggests that the FEAR network is important not just for a timely exit from mitosis but also for the robustness of the time spent in anaphase.

## Predicting cell–cell variability in checkpoint competence of SPoC mutants

Having established that the model can represent and predict temporal effects, we set out to explore how the timing of mitosis can impact the long-term viability of cells. The SPoC delays mitosis, while the spindle is misaligned and mutations to SPoC components allow cells with misaligned spindle to exit mitosis prematurely. However, there are differences between SPoC mutants; *kin4Δ* cells with misaligned spindles spend considerably longer in mitosis than *bub2Δ* cells [46]. SPoC mutants show considerable cell–cell variability in checkpoint competence, with only a fraction of cells exiting mitosis prior to spindle alignment. Precise measurement of the proportion of mutant cells exiting mitosis prior to spindle alignment shows that *bub2Δ* cells are more likely than *kin4Δ* cells to exit mitosis prematurely and become multinucleate [46]. We hypothesised that these differences in timing may explain the differences in outcome between these 2 mutants.

We modified the existing MaBoSS model to allow for the difference in time spent in anaphase for *bub2Δ* and *kin4Δ* cells (Model 6). In this model, the wiring has not changed, but the rate of Tem1 activation is higher in the presence of Lte1. This choice was based on the earlier result that Lte1 inhibits the activity of Bub2-Bfa1 towards Tem1. Note that just as before, this does not necessarily indicate that Lte1 acts as a Guanine nucleotide Exchange Factor (GEF) for Tem1 but may act via a different or even indirect mechanism. To dimensionalise the model, we defined 3 parameters representing the rate of Bfa1 inhibition in the presence ($\rho_{fast}$) or

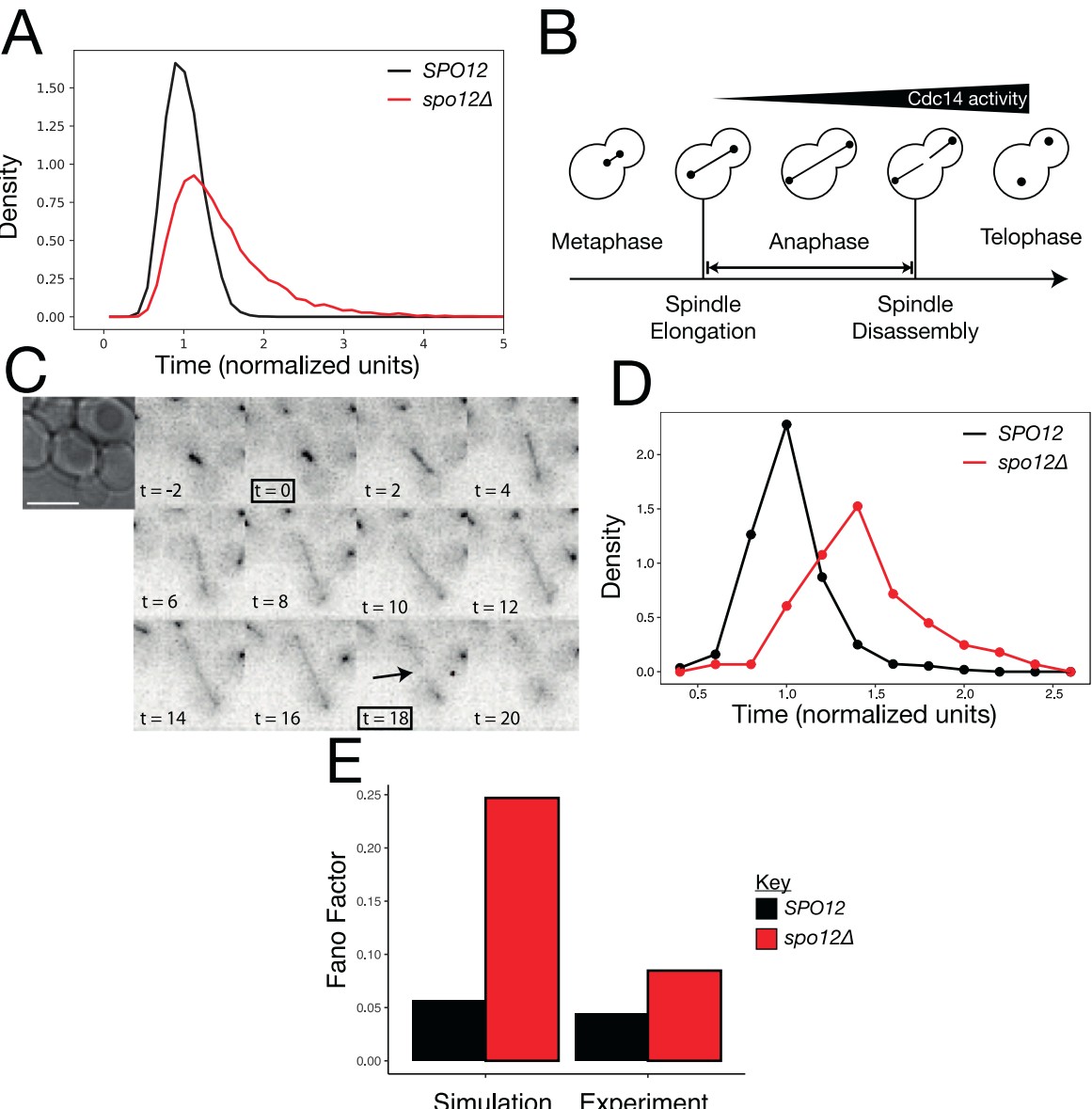

**Fig 6. The role of FEAR in regulating anaphase length.** (A) We simulated 10,000 *SPO12* and *spo12Δ* cells and the length of anaphase (time from model initiation until mitotic exit) was calculated and was normalised to the mean of the wild-type cells. (B) Schematic showing the key cell cycle events used to calculate the length of time spent in anaphase. (C) Time course showing mRuby2-Tub1 fluorescence in a representative cell during exit from mitosis. Images were taken at 2-minute intervals and used to determine the length of anaphase. The image at 0 minutes shows the final frame where the cell has an unextended spindle and spindle disassembly after 18 minutes. (D) Distribution of anaphase lengths in *SPO12* and *spo12Δ* cells. Five time courses were performed, each with 3 fields of view per strain (*SPO12* **n = 281**, *spo12Δ* **n = 223**). Due to differences in mean exit times between time courses, exit times from each time course were normalised to the mean exit time of *SPO12* cells in that time course. (E) The coefficient of variation of exit times for *SPO12* and *spo12Δ* cells in simulation and experiment. Raw data can be found in S7 File; example simulation data and the normalised data can be found in S8 File. FEAR, Cdc14 Early Anaphase Release.

absence ($\rho_{slow}$) of Lte1 and the rate of all other variables ($\rho$) (S1 Table). We chose $\rho$ so that the average length of mitosis in a wild-type cell with an aligned spindle is 25 minutes and $\rho_{slow}$ so that it is 70 minutes for a *kin4Δ* cell (S6A and S6B Fig). We found that varying $\rho_{fast}$ had minimal effect on the length of mitosis in any of the tested mutants so it was left at 1 (S6C Fig). With these parameters, we could simulate cells with accurate temporal resolution, allowing us

to estimate the exit time distributions of both mutant strains (Fig 7A). We define the time taken for a cell to exit mitosis as the random variable, $E$.

In order to determine whether mitotic exit or spindle alignment occurs first, we require an estimate for the time taken to align the spindle. In order to match the findings of Falk and colleagues [46], we decided to model *kar9Δ osTIR1 dyn1-AID* cells [81]. Upon treatment with auxin, these cells become deficient in both of the parallel spindle alignment pathways and so spindle alignment will progress according to the random motion within the cell. As a simplistic model of this process, we consider the spindle to rotate around the centre of the mother compartment, with its angular displacement $x(t)$ behaving as a Brownian motion (Fig 7B). If the SPB ever passes into the bud neck, then we assume the spindle has aligned and cannot become misaligned again. Then, as it does not matter which of the two SPBs eventually enters the bud, we consider $x(t) \in \left( -\frac{\pi}{2}, \frac{\pi}{2} \right)$, with alignment occurring if $x$ passes beyond either $\frac{\pi}{2} - \theta$ or $-\left( \frac{\pi}{2} - \theta \right)$, where $\theta$ is half the angular neck width (Fig 7B and S6E Fig). We assume the orientation of the spindle during metaphase is random, so that the initial value is distributed uniformly $x(0) \sim U\left( -\frac{\pi}{2}, \frac{\pi}{2} \right)$. Example trajectories of $x(t)$ are shown in Fig 7C. We define the random variable, $A$, to be the alignment time, when $x(t)$ first crosses $\pm\left( \frac{\pi}{2} - \theta \right)$. Note that $\mathbb{P}(A = 0) > 0$, as the distribution of initial values includes regions within the zone of alignment, corresponding to the fact that the spindle may be already aligned to the mother–bud axis upon spindle extension. We performed 10,000 simulations of $x(t)$ to generate measurements of $A$. We then used a cubic spline interpolation on the histogram of alignment times to approximate the Probability Distribution Function (PDF) of $A$. We used a similar approach to approximate the PDF of the time until mitotic exit, $E$, from the MaBoSS simulation results (Fig 7D).

With PDFs of $E$ and $A$, we can analytically deduce the distribution of the difference $D = E - A$ as a convolution of the two distributions (Fig 7E)

$$f_D(t) = \int_0^\infty f_E(t+s)f_A(s)ds.$$

Numerically integrating the area between the x-axis, $f_D$ and the line $x = 0$ yields the probability that $D < 0$, which corresponds to the case that mitotic exit occurs prior to spindle alignment, leading to the creation of a multinucleate cell. Note that $E$, but not $A$, depends on the specific SPoC mutation and so must be estimated separately for each mutant. We applied this approach to 10 mutants, where the proportion of successful cell divisions has been determined experimentally by Falk and colleagues [46]. The proportions predicted by our model fit the behaviour measured experimentally (Fig 7F). The only exceptions to this were the reduction in numbers of multinucleate cells caused by the *spo12Δ* mutation in the *LTE1-8N* and *LTE1-8N kin4Δ* backgrounds. Our model predicted a modest reduction in the proportion of multinucleate cells, as a result of the delay caused by loss of FEAR function, while Falk and colleagues [46] measured a more significant reduction. The model also predicts that the SPoC competence of *CDC15-7A MOB1-2A* is comparable to *CDC15-7A* alone, which would be an interesting phenomenon to test experimentally.

Overall, these findings suggest that the compartmental logical framework is capable of representing the continuous properties of the system and can distinguish between "strong" and "weak" SPoC mutants.

## Model predictions

A major strength of the compartmental logical framework is the ability to simulate the impact of mutants that affect localisation independently of activity. We simulated the forced localisation of each of the 10 MEN proteins that localise to the SPB in the model (Fig 8A). Many of these experiments have already been performed, and in these cases, the results mainly agree

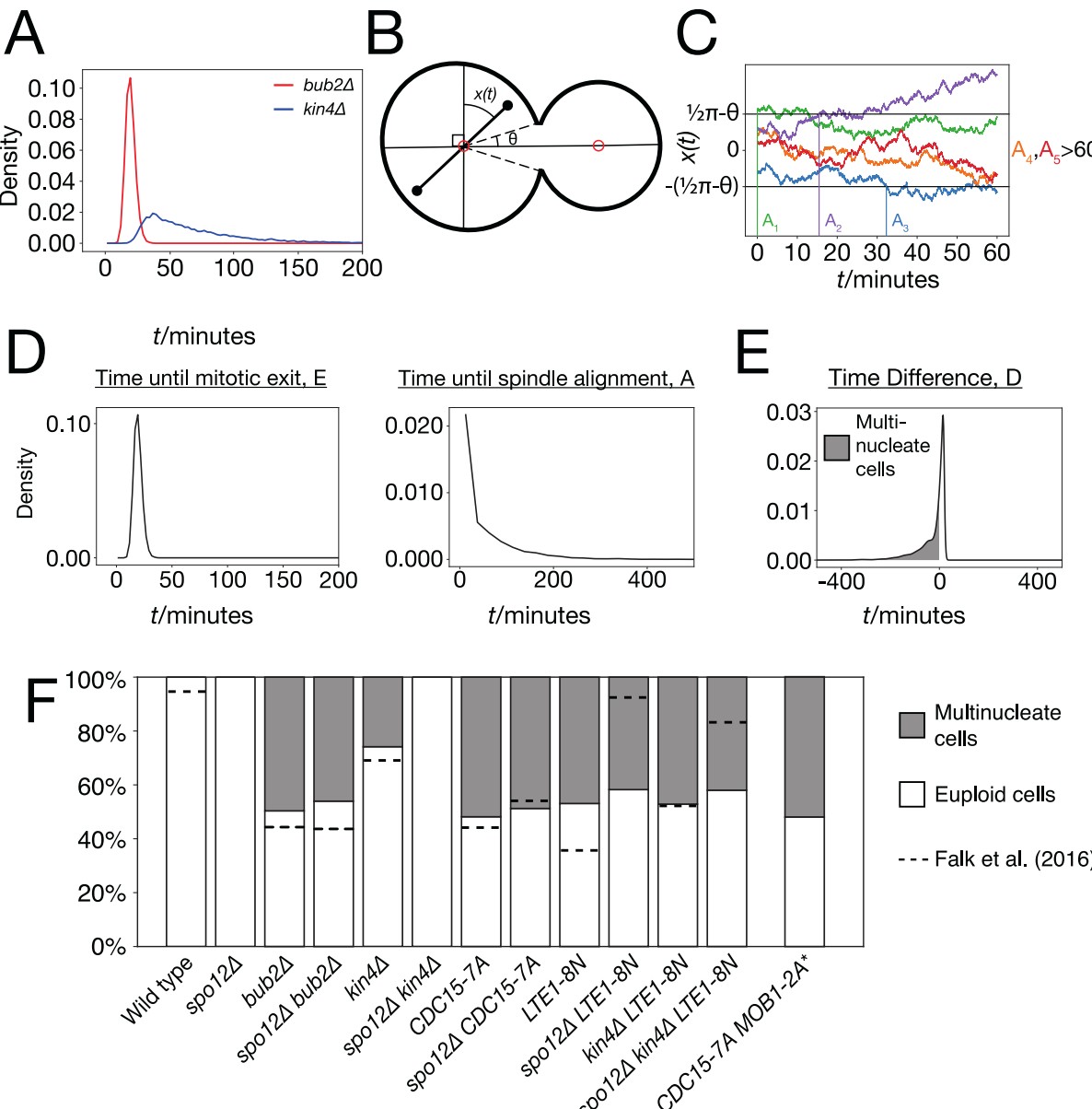

**Fig 7. Use of the parameterised model to predict and explore cell–cell variability in SPoC mutants.** (A) Simulated exit times of *bub2Δ* and *kin4Δ* cells with misaligned spindles, from 10,000 runs of the model. (B) Schematic of a *kar9Δ osTIR1 dyn1-AID* cell, showing the spindle angle *x(t)* and the half-angular neck width, ϑ. (C) Simulations of *x(t)*, the spindle angle starting from uniformly distributed initial conditions and varying as a Brownian motion. The time until alignment $A_i$ is indicated for each simulation. $A_1 = 0$ as in this case, the initial condition of the simulation is within the bud neck ($x(0) > \frac{\pi}{2} - \theta$), corresponding to the scenario where the spindle is aligned at the point of extension. $A_2$ and $A_3$ can be measured as the point where *x(t)* crosses either of the boundaries, as it is not important which SPB enters the bud. The final 2 simulations do not achieve alignment during the 60 minutes simulated so $A_4, A_5 > 60$. (D) The distribution of exit times, E, for a simulated *bub2Δ* mutant and the distribution of alignment times, A, for a simulated *kar9Δ osTir1 dyn1-AID* cell. These distributions were inferred from cubic spline interpolation of histograms generated from 10,000 runs of the model or 10,000 Brownian motion simulations respectively. (E) Distribution of the difference between exit time and alignment time, D, for the simulated *bub2Δ kar9Δ osTir1 dyn1-AID*. The area between the x-axis, the curve, and $x = 0$ gives the predicted probability of a given cell exiting mitosis before spindle alignment occurs, giving rise to a multinucleate cell. (F) Predicted proportions of multinucleate cells for various genetic backgrounds. Dotted lines show the measured proportions of multinucleate cells in Falk and colleagues [46]. *CDC15-7A MOB1-2A* was not included in the assays of Falk and colleagues [46] and so no dotted line is included. Example data can be found in S9 File. SPB, Spindle Pole Body; SPoC, Spindle Position Checkpoint.

with our simulations (5/6). Counterintuitively, Bub2 and Bfa1, which are inhibitors of the MEN, promote mitotic exit in cells with misaligned spindles [70,82], as does Cdc5 [48]. The ability of the compartmental, logical model to reproduce the behaviour of forced localisation of Bub2 and Bfa1 is an improvement over existing compartmental models of the MEN [23], which predict the opposite phenotype (S7 Fig). Our model predicts that forced localisation of Tem1 and Cdc15 promote mitotic exit in all conditions, including in metaphase. The Cdc15 finding is consistent with experimental results [66]. Experiments with a Tem1-Cnm67 fusion protein showed it could promote mitotic exit with a misaligned spindle but not in metaphase [80]. However, it is worth noting that Nud1, rather than Cnm67, is thought to be the scaffold for Tem1 at the SPB [6]. We decided to test whether Tem1 could initiate mitotic exit if forced to interact with Nud1 using the GFP-Binding Protein (GBP) [83]. We expressed *NUD1-GBP* from the reduced strength *GALS* promoter in either wild type, *TEM1-YFP*, or *CDC15-YFP* strains. We found that recruitment of Cdc15 but not Tem1 to the SPB could promote mitotic exit in cells arrested in metaphase (S8A Fig, S10 File). We also found that recruitment of either Tem1 or Cdc15 was lethal to the cell (S8B Fig).

Kin4 is predicted to prevent mitotic exit when forced to localise at the SPB. The experimental evidence for this mutation is conflicted, with a Kin4-Spc72 fusion causing a delay to mitotic exit [77], while a symmetrically localising mutant caused no delay [84]. Kin4 was found to cause a significant growth defect when bound to Spc72 in genome-wide synthetic physical interaction screens [85]. Dbf2, Mob1, and Cdc14 are all predicted to be insufficient to alter MEN signalling at the SPB. Of these, only Mob1 has been tested; it was found that a Mob1-Nud1 fusion was in fact lethal [86], presumably because it prevented movement of Mob1-Dbf2 into the nucleus, an effect not captured by this model. CDK is predicted to behave like Kin4, preventing mitotic exit when forced to localise to the SPB and further this effect is predicted to be rescued by *bfa1Δ* (Fig 8B). A study found that forced localisation of Clb2 to the SPB delayed spindle disassembly [87]; however, no experiments forcing CDK to the SPB have been performed.

We constructed strains expressing *NUD1-GFP* from the endogenous promoter and bearing plasmids expressing either *GBP* or a fusion *CLB2-CDC28-GBP* protein from the *MET3* promoter. We tuned the expression of fusion protein by addition of 10-$\mu$M methionine, to prevent high levels of CDK overexpression [88]. We found that forcible recruitment of CDK to the SPB caused a growth defect (Fig 8C) and that this growth defect was rescued by *bfa1Δ*. Recruiting CDK to the SPB caused cells to arrest in late anaphase, with a large bud and an extended spindle (S8C Fig), and this phenotype was rescued by *bfa1Δ* (S8D Fig, S11 File). This indicates a failure to exit from mitosis, as our model predicts.

Forcible localisation of proteins to both SPBs has been used as a tool to explore the impact of localisation for many years; however, forcing proteins to localise to a single SPB has not been explored in the same detail. An optogenetic binding system has been used to target Clb2 to a single SPB [87]; however, the downstream impact on MEN signalling was not fully investigated. Our compartmental logical model is capable of making predictions of the outcomes of such experiments. Our model predicts that targeting Cdc15 to either of the SPBs would be sufficient to drive mitotic exit (Fig 8D). This suggests that MEN signalling could occur at the mSPB, if Cdc15 was present there. On the other hand, the inhibitory effect of CDK is predicted to occur only when CDK is targeted to the dSPB, with forcible localisation of CDK at the mSPB having no functional effect (Fig 8D).

## Discussion

In this article, we have presented a compartmental, logical model of the control of mitotic exit in yeast. This novel modelling formalism brings together the spatial resolution of

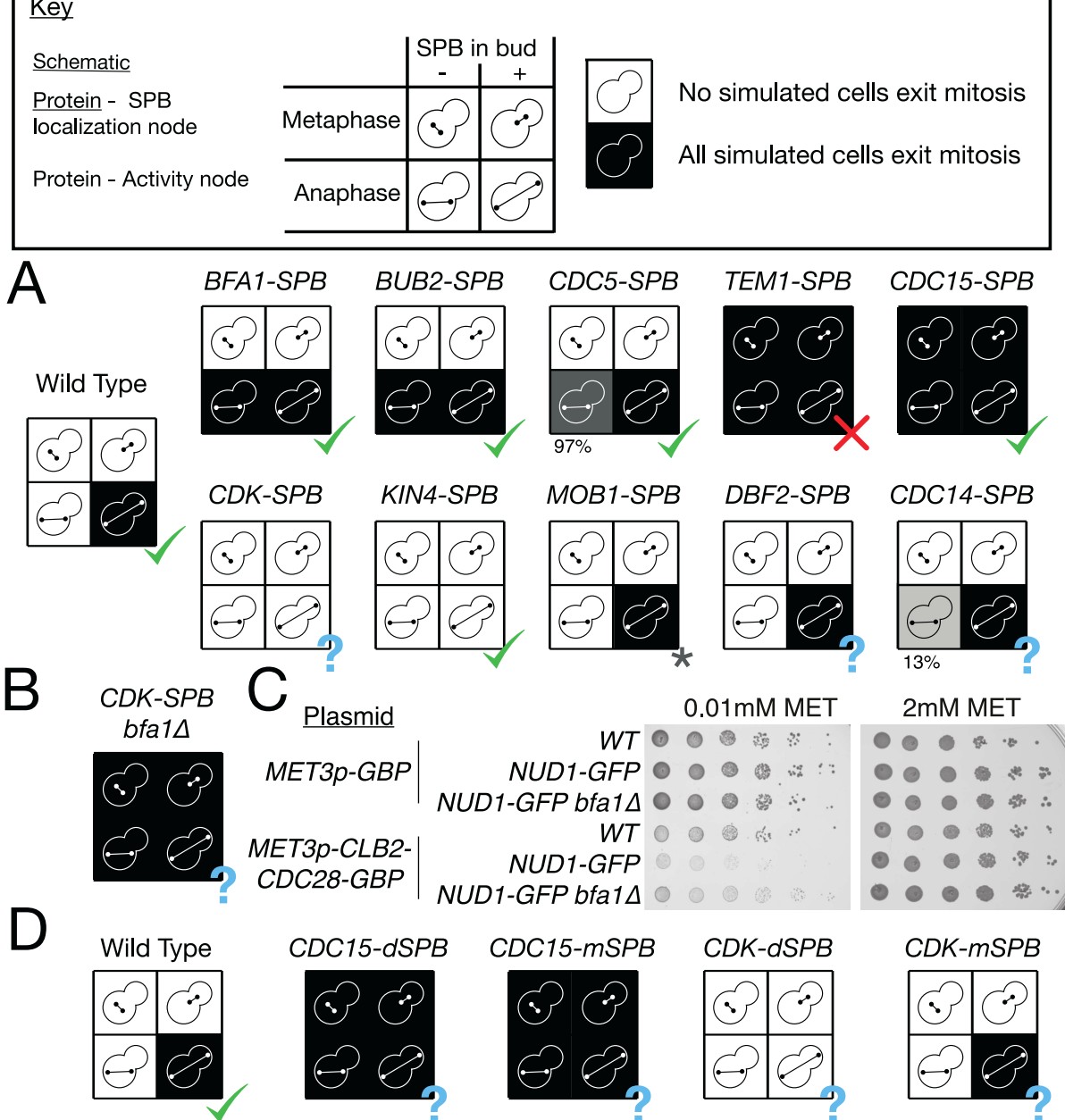

**Fig 8. Forced localisation phenotypes.** (A) Predicted phenotype of cells where each of the SPB-localised proteins in the model are forced to localise to the SPB. A question mark (?) indicates a phenotype that has not been experimentally verified in the literature. A star (*) is used to indicate that the *MOB1-SPB* phenotype differs from literature accounts due to factors beyond the scope of the model. (B) Predicted rescue of the *CDK-SPB* phenotype by *bfa1Δ*. (C) Spot tests showing growth defect of Nud1-GFP cells expressing a fusion Clb2-CDK-GBP protein from the *MET3* promoter and rescue of this defect by *bfa1Δ*. Activity of the *MET3p* promoter was tuned by addition of 10-$\mu$M methionine to media. (D) Predicted phenotype of cells where Cdc15 or CDK are forced to either the mSPB or dSPB. All simulation data can be found in S5 File. CDK, Cyclin-Dependent Kinase; GBP, GFP-Binding Protein; GFP, Green Fluorescent Protein; SPB, Spindle Pole Body; WT, wild type.

compartmental ODE models and the simplicity and scalability of logical models. This model can act as both a representation of our knowledge of the MEN and a tool to guide experimental design; in this article, we have done both.

The choice of the logical formalism comes with inevitable limitations, as reducing the complexities of protein dynamics to discrete levels of activity is a significant simplification. In

some cases, such as the treatment of Kin4 regulation of Bub2-Bfa1, we were able to simplify a quantitative mechanism, in this case a change of turnover rate, into a simpler mechanism whereby Kin4 simply keeps Bub2-Bfa1 off the SPB. In other cases, the model has not been able to represent effects such as low-level localisation of MEN proteins in metaphase [11,74] or symmetric localisation of MEN proteins late in anaphase [89]. The implicit representation of complexes in this model also represents a further limitation. It is possible these limitations could be overcome, for example, by addition of extra levels for these nodes; however, it seems likely that some mechanisms will never be fully captured by logical rules. We argue that despite these limitations, the simplicity of the formalism makes it a useful tool to explore networks like the MEN.

The steps taken to optimise and refine the model provide some insight into key aspects of MEN regulation. We included 2 levels of Bub2-Bfa1 and Tem1 activation in order to accurately model the effect of *bub2Δ* or *bfa1Δ* mutations. The necessity of this step shows the importance of Bub2-Bfa1 to tune the strength of MEN response throughout mitosis. In order for *kin4Δ spo12Δ* cells to maintain a SPoC, we required 2 parallel Bub2-Bfa1 regulation pathways (Fig 9). This mechanism was first proposed by Falk and colleagues [46], but the molecular details are still unclear. Our modelling suggests that Lte1 prevents inhibition of Tem1 by Bub2-Bfa1, but models targeting either Tem1 or Bfa1 were equally effective at explaining the data. Model 6 proposed that Lte1 activity is also important for the timing of MEN signalling in the presence of Bub2-Bfa1. Lte1 has long been known to contain a GEF-like domain; however, no GEF activity towards Tem1 could be detected in vitro [78]. Therefore, either Lte1's Kin4-independent activity towards Bub2-Bfa1 acts through an as yet unknown intermediate protein or Lte1's GEF activity depends on factors not included in the reactions of Geymonat and colleagues [78], for example, scaffolding by SPB components. While experiments show the *lte1Δ kin4Δ spo12Δ* triple mutant struggles to exit mitosis, it is still able to do so with low efficiency [48], demonstrating that other bud-localised proteins, such as Ste20, can promote MEN activity. Indeed, this model uses a coarse-grained representation of polarity proteins, and the model could be expanded to represent known interactions of the MEN with Ste20, Kel1 and 2, Cdc24, and Cdc42 [90]. The *lte1Δ kin4Δ spo12Δ* mutation prevents exit from mitosis in Model 4, as Ste20 is not currently represented in the model, although this could be implemented in future versions. Finally, in order to fit the phenotype of *CDC15-7A MOB1-2A* cells, we introduced an ASC responsible for controlling Cdc15 localisation in the absence of Tem1 and CDK. Integrating the findings of Rock and colleagues [66] and Botchkarev and colleagues [72], we propose that the movement of Cdc5 across the nuclear membrane to the cytoplasmic face of the SPB in anaphase is the signal represented by the ASC (Fig 9). Botchkarev and colleagues [72] propose that the translocation of Cdc5 is controlled by the FEAR network, and this could be tested in future models.

It is interesting to note that in this model, the essential role of Cdc5 is in the localisation of Cdc15 and not in inhibition of Bfa1. Certainly, if we accept that Cdc5 is restricted to the nucleus in metaphase and that *CDC15-7A MOB1-2A* cells have a SPoC in metaphase, we must accept that Bfa1 can become inhibited without Cdc5. This view is supported by the findings of Rock and colleagues [66] that Mob1-Dbf2 can be activated in the absence of Cdc5 and Tem1, when Cdc15 is artificially localised at the SPB. There is some disagreement in the literature over whether deletion of *BUB2* or *BFA1* can overcome the effects of Cdc5 inactivation. While some studies have found that the lethality of *cdc5-1* [76] and *cdc5-10* [12] can be totally reversed by deletion of *BUB2* or *BFA1*, other studies show these deletions cannot fully rescue *cdc5-2*, especially at 37˚C [75,91,92]. It is difficult to dissect the exact roles of Cdc5 because of the many roles it plays in mitosis, meaning these different temperature-sensitive alleles are probably defective in slightly different functions. However, it seems clear that Cdc5 has other

essential roles than just in Bub2-Bfa1 regulation. It is also worth noting that a *bfa1-11A* allele in which Cdc5 sites were mutated did not result in a mitotic arrest [75]. The model makes a number of testable predictions relating to rescue of Cdc5 mutants (S9 Fig). The model predicts that simultaneous deletion of *CDC5* and *BUB2* or *BFA1* would be able to exit mitosis. This is because the model predicts that the hyperactive Tem1 in these strains would be able to recruit Cdc15 even in the absence of Cdc5. It also recapitulates the finding of Rock and colleagues [66] that recruitment of Cdc15 to the SPB can overcome the effects of *CDC5* disruption. Furthermore, it predicts that a *CDC15-7A cdc5*Δkin4Δspo12Δ strain would not only be viable but also would recover function of the SPoC. It seems clear that the pleiotropic nature of Cdc5 and the variable defects of different temperature-sensitive *CDC5* alleles have made it difficult to precisely unravel the contribution of Cdc5 to mitotic exit. Further experiments utilising conditional systems to deplete Cdc5 activity or phosphomimetic mutations will help to test the predictions of the model and clarify the role of polo kinase.

Our model predicted that the FEAR network is important not just for timely mitotic exit but also to even out variation in the length of mitosis (Fig 9). This finding was validated by experimental measurements of the length of anaphase in wild-type and FEAR mutant cells. While the FEAR mutants have long been known to interact genetically with MEN mutants, the purpose of the network, beyond speeding up anaphase, has been difficult to understand. Our solution is that FEAR is primarily a mechanism to limit the variability of the length of mitosis. In our model, it is the activity of FEAR towards Cdc15 which determines anaphase variability; however, other aspects of regulation may play a role. For example, Cdc14 reverses CDK phosphorylation of securin, creating a positive feedback loop in the FEAR and sharpening the metaphase–anaphase transition [93]. Future versions of the model could be updated to include this. It is interesting to note that single cell measurements of cell cycle stages in human cells show they follow an Erlang distribution, which represents the time taken by $k$ independent Poisson processes of rate $\lambda$ [94,95]. There is a direct parallel here to the continuous-time Markov chain used by MaBoSS, in which the update times for each node are independent Poisson processes. This suggests that this implementation of the logical modelling framework will be an effective tool to model the length of cell cycle stages.

We have proposed a link between the speed of MEN activation and the strength of SPoC mutants, which is capable of reproducing the behaviour of a number of mutants. Furthermore, we found that after fitting rate parameters to the logical model, we were able to fit the percentage of multinucleate cells in both *kin4Δ* and *bub2Δ* by fitting a single parameter, $\sigma$. This suggests the model is not overfitted to the data. This demonstrates how the qualitative logical approach can be easily adapted to make quantitative predictions. Our approach, based on the difference in timing between checkpoint satisfaction and signalling, is likely to apply more broadly to the study of other checkpoints which have mutants of varying strengths.

Our model correctly predicted the phenotype caused by forcing most MEN proteins to localise at the SPBs. This demonstrates that the compartmental logical framework is versatile enough to make predictions about the impact of protein mislocalisation. Protein mislocalisation is a powerful tool to probe protein function [96], and compartmental logical models may aid in guiding design of these experiments. Synthetic physical interaction screens with the SPB found an enrichment for MEN proteins among proteins which cause a growth defect when forced to interact with Nud1 [85], as well as identifying mitotic regulators such as Glc7 (PP1) and Tpd3 (PP2A). The model incorrectly predicts that forcing Tem1 to the SPB can initiate mitotic exit in metaphase, as demonstrated by the experiments of both Valerio-Santiago and Monje-Casas [80] and ourselves. The *bub2Δ* or *bfa1Δ* mutations cause mitotic exit to occur in metaphase by promoting premature Tem1 loading, so exploring why this mutation but not *TEM1-SPB* can have this effect will be an interesting direction for further experimentation. It

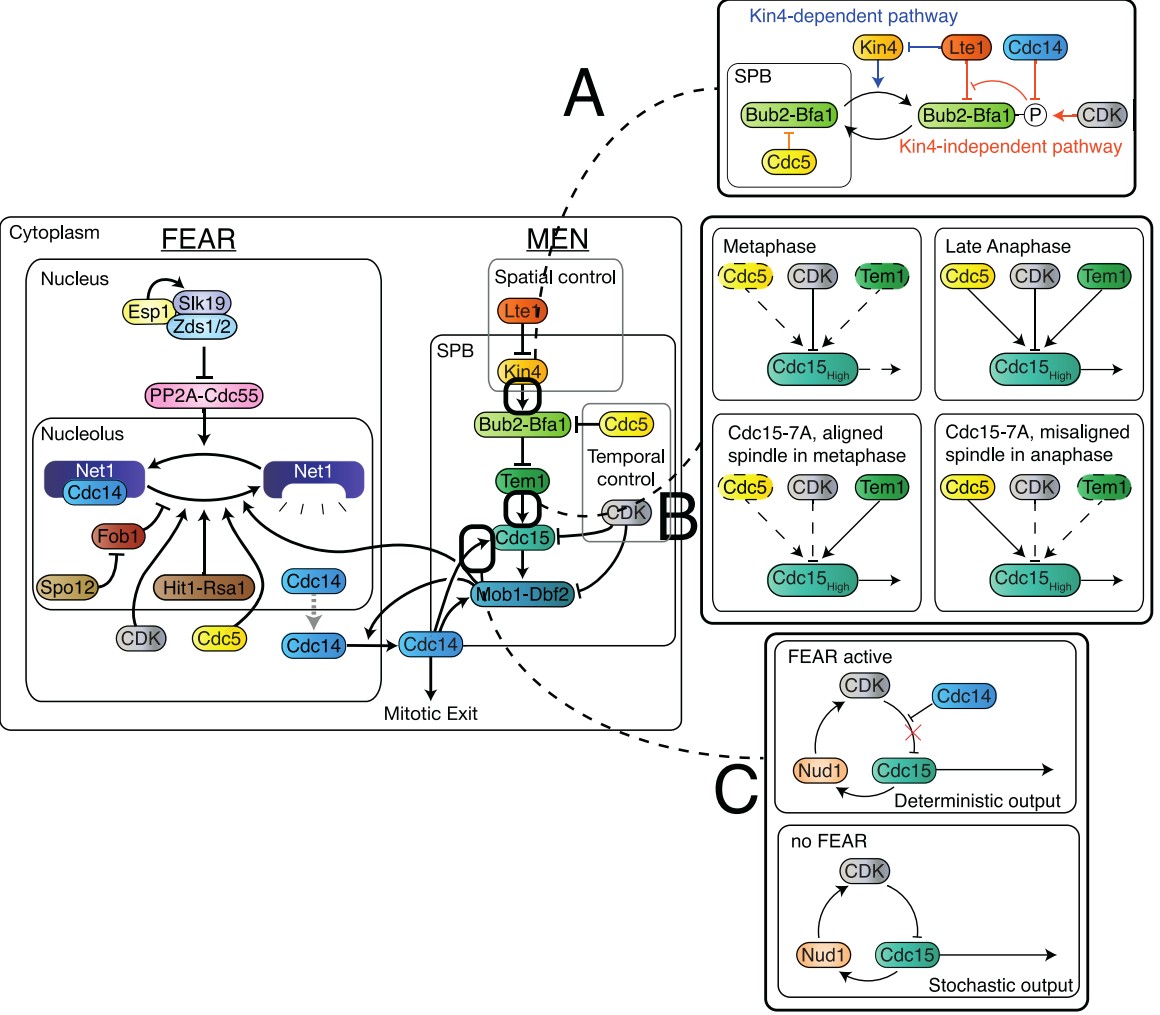

**Fig 9. Model of the MEN including the developments contributed in this manuscript.** (A) Lte1 regulates Bub2-Bfa1 via 2 pathways, only one of which is dependent on Kin4. (B) Cdc5 is required for recruitment of Cdc15 to the SPB in the absence of Tem1 or CDK regulation. (C) FEAR breaks a Cdc15-Nud1-CDK negative feedback loop, leading to deterministic timing of mitosis. CDK, Cyclin-Dependent Kinase; FEAR, Cdc14 Early Anaphase Release; MEN, Mitotic Exit Network; SPB, Spindle Pole Body.

is also interesting to note that some mutations that lose control of mitotic exit have no impact on viability (such as bub2Δ), while perturbations such as Cdc15-SPB are lethal.

While many experiments have been performed forcing protein localisation at both SPBs, none have explored the impact of forcing MEN proteins to a single SPB. Our model predicts that the MEN could be activated in metaphase or in cells with misaligned spindles if Cdc15 is targeted to either SPB. This could result in the reversal of the usual asymmetry, with MEN activity occurring at the mSPB rather than the dSPB. On the other hand, our model predicts that the inhibitory effect of forced CDK localisation would occur only when CDK is targeted to the dSPB, with no effect resulting from forced localisation at the mSPB. It will be an important test of the model to conduct these experiments, which could be performed with an optogenetic binding system such as Yang and colleagues [87].

Our model of the MEN can represent approximately 80% of the genetic phenotypes we tested in the validation stage, for a total of 165/199 phenotypes matching experiments when combined with the FEAR network training set. These phenotypes include deletions,

temperature-sensitive alleles, overexpressions, and most importantly for this study, localisation mutants. This success demonstrates how the logical modelling framework is capable of capturing many of the essential aspects of MEN regulation. However, there are still some phenotypes the model cannot predict, and in particular, the model struggles to represent overexpression and temperature-sensitive alleles. Overexpression is a complex genetic perturbation [38], and as we demonstrate with the examples of *GAL1p-CDC15 tem1Δ* and *GAL1p-KIN4 rts1Δ*, a single set of rules for how overexpression interact with other aspects of regulation is not possible. Looking forward, we must either accept the low level of inaccuracies caused by blanket treatments or find ways to integrate further quantitative details about how proteins interact in order to make more precise predictions about how overexpression may interact with other mutations. Temperature sensitivity is also a difficult perturbation to model, with the model treating such mutants as identical to full deletions. Dealing with this limitation will be particularly important if genome-scale models are to be benchmarked against systematic genetic interaction datasets such as Costanzo and colleagues [97]. Existing approaches to prediction of genetic interactions from logical models still do not allow for partial loss of function mutations, for example, Calzone and colleagues [98]. If a methodology to understand the impact of partial loss of function mutations could be developed, this would open the doors to the use of logical models to predict phenotype from genotype at the genome scale, with important applications to personalised medicine.

## Supporting information

**S1 Fig.** (A) Activity and localisation networks underlying the compartmental model. (B) Schematic showing how logic gates are expanded to include overexpression (OE) nodes and the corresponding logic gate.
(EPS)

**S2 Fig. In Model 4, Lte1 directly inhibits Bfa1 activity, leading to correct prediction of the phenotype of *GAL1p-KIN4* cells.** A version where Lte1 directly activates Tem1 (Model 4a) predicts behaviour of *GAL1p-KIN4* incorrectly. An alternative version (Model 4b) where Lte1 targets Tem1 activity and not localisation is also able to fit the phenotypes that Model 4 can. All simulation data can be found in S5 File.
(EPS)

**S3 Fig. A model (5a) in which Cdc5 localises to SPBs as a results of interaction with the high level of Bub2-Bfa1 fails to capture the phenotype of *CDC15-7A MOB1-2A* cells.** Identifying the ASC as Cdc5 in Model 3 (Model 3a) leads to incorrect behaviour of *CDC15-7A MOB1-2A*. Model 3 requires an additional pathway linking Lte1 to Tem1. All simulation data can be found in S5 File.
(EPS)

**S4 Fig. Representative images of T744 and T747 cells, showing both mRuby2-Tub1 and Cdc14-CFP fluorescence.** Scale bars show 5 $\mu$m.
(EPS)

**S5 Fig.** (A) Raw measurements of anaphase length, grouped by strain and time course. Box plots show the median and the upper and lower quartiles, whiskers show 1.5 times the interquartile range or the closest measurement, whichever is closest. The raw data can be found in S7 File. (B) Log density distribution of simulated exit times for *SPO12* and *spo12Δ* cells. The *spo12Δ* distribution show a power law tail, while the *SPO12* distribution does not. (C) Log density distribution of experimentally observed exit times for *SPO12* and *spo12Δ* cells.

Both distributions show a power law tail; however, the gradient of the *spo12Δ* distribution is shallower. The data used to plot panels B and C can be found in S8 File.
(EPS)

**S6 Fig. Parameter selection for the dimensional model (Model 6).** (A) The basic rate, $\rho$, was chosen so that the mean of the exit time distribution of wild-type cells is 25. We simulated 10,000 anaphase cells with aligned spindles for 40 values of $\rho$ between 0.6 and 1.0 and calculated the mean exit time. The closest to the target value ($\rho$ =0.84) was selected. (B) The slow rate of Bfa1 inhibition, $\rho_{slow}$, was chosen so that the mean of the exit time distribution of *kin4Δ* is 70. We simulated 10,000 anaphase cells with misaligned spindles for 18 values of $\rho_{slow}$ between 0.004 and 0.018 and calculated the mean exit time. The closest to the target value ($\rho_{slow}$ = 0.012) was selected. (C) We tried varying the fast rate of Bfa1 inhibition, $\rho_{fast}$ over 2 orders of magnitude but found it had little effect on the length of mitosis in either mutant, so it was left at $\rho_{fast}$ = 1. Mean exit times were derived from simulations of 10,000 anaphase cells with misaligned spindles. (D) The parameter, $\sigma$, representing the rate of spindle alignment, was chosen to match both the measured proportions of multinucleate cell formation in *bub2Δ* (approximately 0.5) and *kin4Δ* (approximately 0.25). We tested 6 values of $\sigma$ between 0.11 and 0.16. Fortunately, the value $\sigma$ = 0.14 fits both proportions closely. Mean exit times were derived from simulations of 10,000 anaphase cells with misaligned spindles. (E) Measurement of the half-angular bud width, $\vartheta$, from a microscope image of a large-budded wild-type cell. Based on this measurement, we use a value of $\theta$ = 0.3. Example simulation results can be found in S12 File.
(EPS)

**S7 Fig. Simulations of the model of [23].** In this model, activation of the MEN is signalled by the number of Tem1-GTP molecules exceeding 65 (the MEN threshold). Simulations were performed using the same parameters as [23], with custom initial conditions matching the pre-alignment steady states of the model. The simulation data can be found in S13 File.
(EPS)

**S8 Fig.** (A) Recruiting Cdc15 but not Tem1 to the SPB promotes mitotic exit in metaphase. Wild-type, *TEM1-YFP*, and *CDC15-YFP* cells expressing *NUD1-GBP* from a plasmid were synchronised with alpha factor and then arrested in metaphase with nocodazole. After 3 hours, the number of single and multi-budded cells was counted. Error bars represent 95% confidence intervals calculated with the Clopper–Pearson method. *P* values were calculated using the 2-tailed Fisher exact test. The data for this plot can be found in S8 File. (B) Forced interaction of both Tem1 and Cdc15 with Nud1 is lethal. (C) Representative images of *NUD1-GFP MET3p-CLB2-CDC28-GBP-RFP* cells, grown in media containing 10-$\mu$M methionine. We placed cells into 4 categories: G1, pre-anaphase spindle (S or early M cells with 2 SPBs less than 3 $\mu$m apart), anaphase (SPBs over 3 $\mu$m apart), and abnormal cells (aberrant SPB or bud number). (D) Quantification of the percentage of cells in each category. *NUD1-GFP MET3p-CLB2-CDC28-GBP-RFP* cells showed a high proportion of anaphase cells, which could be rescued by repression of the *MET3* promoter by addition of methionine or by the *bfa1Δ* mutation. *MET3p* activity was tuned by addition of 0.01 mM methionine (+) or 2 mM methionine (−). *P* values calculated using the 2-tailed Fisher exact test. The data for this plot can be found in S9 File.
(EPS)

**S9 Fig. The model predicts that the lethality of *cdc5Δ* can be surpressed by *bub2Δ*, *bfa1Δ*, *CDC15-SPB*, or *CDC15-7A*.**
(EPS)

**S1 Video. Time-lapse video showing anaphase in 2 *spo12*Δ cells, arrows indicate the time and location of spindle breakage.** Spindles are visualised with an mRuby2-Tub1 marker (left), and cell morphology is imaged using bright-field microscopy (right). The time between the first frame of spindle extension and the first frame of spindle disassembly (arrows) is used to determine anaphase length.
(MP4)

**S1 Table. Parameters used to simulate SPoC compentence.**
(PDF)

**S2 Table. Plasmids used in this study.**
(PDF)

**S3 Table. Strains used in this study.**
(PDF)

**S1 File. Initial conditions used to simulate the logical model.**
(XLSX)

**S2 File. Prior Knowledge Network used to train the FEAR network, with sources in literature.** Edges go from "Interactor 1" to "Interactor 2" with activation/inhibition specified by the "sign" (+/−). The iterature used to justify each edge is also included.
(XLSX)

**S3 File. Literature phenotypes used to train FEAR network.** The mutant is defined by perturbations of up to 5 proteins. The phenotype is defined by the cycle stage in question: either metaphase ("Metaphase?" = 1) or anaphase ("Metaphase?" = 0), and whether Cdc14 is released at this stage. Addtional information on the exact mutant and the literature reference are also included.
(XLSX)

**S4 File. Rules of the activity and localisation networks in Model 5.** "Rule" gives the formal specification used to build the model using symbolic logic, and "Description" is a brief description of the molecular basis for the given rule. Literature references to justify these rules are also included.
(XLSX)

**S5 File. Simulation results used in various figures.** Exits show the number of simulated cells exiting mitosis, timeouts show the number where the simulation timed out before reaching a steady state or exiting mitosis.
(XLSX)

**S6 File. Literature phenotypes used to validate the model.** The mutant is defined by perturbations of up to 5 proteins. The phenotype is defined by the cycle stage in question: either metaphase ("Metaphase?" = 1) or anaphase ("Metaphase?" = 0) and spindle aligned (Spindle Aligned? = 1) or misaligned (Spindle Aligned? = 0), and whether Mitotic Exit occurs at this stage. Addtional information on the exact mutant and the literature reference are also included. The column "Simulated mitotic exits" states the number of simulated cells (out of 100) that exited mitosis. The column "Simulated timeouts" states the number of simulated cells (out of 100) that timed out before either a steady state was reached or mitotic exit occurred. The column "Correct?" states whether the simulated phenotype matches the one

described in the literature.
(XLSX)

**S7 File. This table summarises the data from live cell time courses used to establish anaphase length.** "Strain" shows which strain of yeast is being examined, "View" indicates separate fields of view, "Cell" indicates separate cells within each file of view, "Last metaphase frame" indicates the last time point at which the cells were judged to be in metaphase, "Complete?" indicates whether cells did (1) or did not (0) break down their spindle during the imaging period, "spindle breakdown" indicates the time point at which cells no longer contained an intact spindle, and "Time" indicates the total time taken from metaphase to spindle breakdown. To the right, the raw mean and variance of the "Time" are shown.
(XLSX)

**S8 File. This file contains data used in Fig 6.** The first sheet contains the distribution of anaphase lengths calculated from MaBoSS simulations of Model 5. Note that as this is constructed from the output of a stochastic algorithm, there may be some minor difference between the example data in the file and the plotted values. The second sheet contains the distribution of anaphase lengths calculated from normalised measurements of anaphase length by time-lapse microscopy. The third sheet contains the Fano factors of both the simulated and experimentally derived distributions.
(XLSX)

**S9 File. This file contains data used in Fig 7.** The first sheet contains the distribution of exit times prior to spindle alignment in 2 SPoC mutants (*bub2Δ* and *kin4Δ*) calculated from MaBoSS simulations of Model 6. The second sheet contains example trajectories of the simulated spindle angle in *kar9Δ dyn1-AID osTIR1* cells, modelled as a Brownian motion. The third sheet contains the distribution of alignment times in *kar9Δ dyn1-AID osTIR1* cells calculated from Brownian motion simulations. The fourth sheet contains the distribution of the difference between the spindle alignment and exit times for a *bub2Δ* mutant prior to spindle alignment, calculated from MaBoSS simulation of model 6 and Brownian motion simulations of spindle alignment. The fifth sheet contains the predicted percentage of binucleate cell phenotype after 1 cell division, calculated from MaBoSS simulation of Model 6 and Brownian motion simulations of spindle alignment, for various mutants.
(XLSX)

**S10 File. This table summarises the data from live cell imaging of cells expressing *NUD1-GBP*, arrested with nocodazole.** "Strain" refers to the strain of yeast being imaged (E438 is a wild-type control), "View" refers to the field of view, "Large budded" is the number of cells with a bud roughly equal in size to the mother, "Small budded" is the number of cells with a bud smaller than the mother, "Multi-bud" is the number of cells with more than one bud, and "Total" refers to the total number of cells (including unbudded cells) visible. The proportion of multi-budded cells in the entire population and among budded cells was calculated for the total from each strain.
(XLSX)

**S11 File. This table summarises the data from live cell imaging of cells expressing CDK-GBP, from MET3p.** "Number" columns shows the number of cells of each strain, and each methionine concentration were counted in each cell cycle stage; "Percentage" columns show the percentage of cells in each cell cycle stage.
(XLSX)

**S12 File. This file contains data used to plot S6 Fig.** The first sheet contains example data showing mean length of anaphase in simulated wild-type cells for differing values of $\rho$. The second sheet contains example data showing mean length of anaphase in simulated *kin4Δ* cells with misaligned spindles for differing values of $\rho_{slow}$. The third sheet contains example data showing mean length of anaphase in simulated mutant cells for values of $\rho_{fast}$ over 2 orders of magnitude. The fourth sheet contains example data showing percentage of binucleate cells in simulated mutant cells for differing values of $\sigma$.
(XLSX)

**S13 File. This file contains data used to plot S7 Fig.** The first sheet contains the molecules of (active) Tem1-GTP at the SPB in simulations of wild-type, Bfa1-SPB, and *bfa1Δ* cells as calculated from the ODE model of Caydasi and colleagues (2012). Spindle alignment occurs at t = 1800 s. The second sheet shows the same data for Bfa1 at the SPB.
(XLSX)

**S1 Text.** List of abbreviations, detailed descriptions of the FEAR network and MEN, and explanation of the scope of the model.
(PDF)

## Acknowledgments

We thank V. Noël and L. Calzone for support with the MaBoSS package and A. Gábor and J. Saez-Rodriguez for support with the CellNOptR package. For strains, comments, and suggestions, we thank G. Ólafsson, M. Geymonat, F. Caudron, S. Santos, P. Bates, W. Taylor, U. Eggert, J. Diffley, and the Scientific Computing STP (The Francis Crick Institute).

## Author Contributions

**Conceptualization:** Rowan S. M. Howell, Peter H. Thorpe, Attila Csikász-Nagy.

**Investigation:** Rowan S. M. Howell, Cinzia Klemm, Peter H. Thorpe, Attila Csikász-Nagy.

**Supervision:** Peter H. Thorpe, Attila Csikász-Nagy.

**Validation:** Rowan S. M. Howell, Cinzia Klemm.

**Writing – original draft:** Rowan S. M. Howell, Peter H. Thorpe, Attila Csikász-Nagy.

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
