## [Editor Report · Decision Letter 0]

30 Apr 2020

Dear Dr Csikász-Nagy, 

Thank you for submitting your manuscript entitled "Unifying the mechanism of mitotic exit control in a spatio-temporal logical model" for consideration as a Research Article by PLOS Biology.

Your manuscript has now been evaluated by the PLOS Biology editorial staff as well as by an academic editor with relevant expertise and I am writing to let you know that we would like to send your submission out for external peer review.

Please re-submit your manuscript within two working days, i.e. by May 02 2020 11:59PM.

Kind regards,

Di Jiang,

Associate Editor

PLOS Biology

---

## [Decision Letter · Decision Letter 1]

15 Jun 2020

Dear Dr Csikász-Nagy,

Thank you very much for submitting your manuscript "Unifying the mechanism of mitotic exit control in a spatio-temporal logical model" for consideration as a Research Article at PLOS Biology. Your manuscript has been evaluated by the PLOS Biology editors, an Academic Editor with relevant expertise, and by three independent reviewers.

In light of the reviews (below), we will not be able to accept the current version of the manuscript, but we would welcome re-submission of a much-revised version that takes into account the reviewers' comments. We cannot make any decision about publication until we have seen the revised manuscript and your response to the reviewers' comments. Your revised manuscript is also likely to be sent for further evaluation by the reviewers.

We expect to receive your revised manuscript within 2 months. 

**IMPORTANT - SUBMITTING YOUR REVISION**

*Re-submission Checklist*

*Published Peer Review*

*PLOS Data Policy*

*Blot and Gel Data Policy*

Sincerely,

Di Jiang, PhD

PLOS Biology

REVIEWS:

Reviewer #1: In the manuscript entitled "Unifying the mechanism of mitotic exit control in a spatio-temporal logical model" the authors aim to provide a full mathematical model based on a novel compartmental logical modelling framework that can represent spatial regulation of the mitotic exit network (MEN) proteins. As the authors claimed the model is able to reproduce the phenotype of around 80% of published data/mutants, indicating that despite the difficulty of the network and the lack of some mechanistic details, the model can predict pretty well mitotic exit. This allowed the authors to extract some nice conclusions like the idea that FEAR is required for the regulation of the time spent in anaphase and provide some experimental data to support this point. However, some essential points in the regulation of mitotic exit are missing:

1. The model takes into account as compartments the nucleus, cytoplasm, bud, mSPB and dSPB. I find strange that the authors do not include the nucleolus as a different spatial compartment since the Cdc14 activation depends on its release from the nucleolus to the nucleus. In order to properly model FEAR-Cdc14 release to distinguish among nucleus and nucleolus is required. 

2. In line 276 the authors state "however the activity of Cdc5 is thought to be stable throughout late mitosis, suggesting it is not part of the temporal signal initiating FEAR release." This is incorrect. It has been demonstrated that Cdc5 activation occurs as consequence of Cdc28-Clb2 phosphorylation (Mortensen et al, 2005; Rodriguez-Rodriguez et al, 2016) and that sequential activation of Cdc5 by a second Cdk1 phosphorylation is required for Cdc5 activity in late mitosis. Moreover, it was described that Cdc5 interacts with the FEAR component, separase (Rahal and Amon, 2008) and to contribute to the FEAR-Cdc14 nucleolus release (Shou et al., 2002; Visintin et al., 2003).

3. On page 11, the authors indicated that "restriction of mitotic exit to anaphase posed a challenge in the model". The fact that they missed the sequential activation of Cdc5 (see point 2) as other recently published FEAR-related proteins such as Hit1 (Santos-Velazquez et al 2017) and PP2A-Cdc55 role regulating MEN proteins (see points 6-7) limited their model on this aspect.

4. The authors propose that their model predict a role of Cdc5 in Cdc15 localization and Lte1 regulation of Bfa1 (as stated in the abstract and in their model). However, they do not provide any experimental data supporting these findings (although they provide some experimental data supporting other points). 

5. The model failed mostly in reproducing overexpression experiments. The authors demonstrated that Cdc5 OE is not able to rescue mob1 deleted mutant as the model predicted (contrary to the observed results in thermosensitive mutants). However, since their model predicts Cdc14 activation as an output of mitotic exit activation, they should check Cdc14 localization and release instead of rescue on plates in the mob1 deleted mutant, since other MEN-regulated proteins could be affected in this experiment (i.e. cytokinesis regulation by MEN).

In the published Cdc5 OE experiments in a dbf2-2 ts mutant, Cdc14 is released only at the nucleus (not into the cytoplasm) and failed to sustain Net1 hyperphosphorylation as MEN is not active (Rodriguez-Rodriguez et al 2016). This "partial" release and intermediate phenotype on Net1 phosphorylation are not clear to be depicted in the model.

6. Model 5 include a modification of the rule for Bfa1 to ensure that it can be inhibited at SPB in metaphase. The authors stated that "in this version Cdc5 is only essential to recruit Cdc15 and not in Bfa1 inhibition". This is a bit odd since regulation of Bfa1 phosphorylation is an essential step in MEN regulation and the convergent point of many checkpoint regulations (DDC, SPOC and SAC, reviewed in Matellan and Monje-Casas 2020). Moreover, the authors propose a two-step regulation on Bfa1, Kin4 dependent and Kin4 independent. As previously mentioned, the model did not include the Bfa1 dephosphorylation by PP2A-Cdc55 (Baro et al 2013) and including this step might remove the necessity for this second step on Bfa1 regulation.

7. In addition, since PP2A-Cdc55 activity is in turn control by FEAR (Esp1), this will provide an additional MEN dependent regulation by FEAR, imposing a temporal order and delaying MEN activation until anaphase. Therefore, including these regulations could help to reproduce the temporal order of the FEAR and MEN activation during mitosis. 

Minor points.

1. The abstract could be simplified and restructured to focus in the main findings demonstrated by the authors, like the deterministic timing of anaphase imposed by FEAR.

2. The authors do not make it clear what is the output of the model, Cdc14 full release and activation or mitotic exit (entering into new G1). 

3. Supplementary excel S6, column w is not in English

4. The use of many abbreviations and acronyms in supplementary files makes them difficult to follow. 

5. In Fig.6 the authors used a Cdc14-CFP strain but did not report the results in the Cdc14 release during anaphase. Including this data, together to the spindle elongation will help to see whether the model and the experimental data follow the same pattern on Cdc14 activation. 

6. Line 459 misspelling of dephosphorylation

7. I am not an expert in computational biology, and I cannot assess whether the computational methods and packages used are correct or the most adequate.

Reviewer #2 (Xiaoxue Snow Zhou and Angelika Amon, signed review): This work represents an interesting new theoretical model of mitotic exit in budding yeast and deserves publication in PLOS Biology. In what follows we make several suggestions in points the authors may want to consider before publication:

In this manuscript, Howell et al. describes a compartmental logical model of the mitotic exit network (MEN) which controls exit from mitosis in budding yeast. The MEN is a GTPase-kinases signaling cascade scaffolded onto the outer plaque of the spindle pole bodies (SPBs) to sense the spindle position during mitosis. To ensure accurate genome partitioning, the MEN integrates both the spatial cue of spindle position through the MEN GTPase Tem1 and temporal cue of cell cycle progression through the downstream MEN kinases Cdc15 and Dbf2-Mob1. In addition to protein activities, localization to the SPBs is an important aspect of regulation for MEN proteins. Although the MEN has been studied extensively, a mathematical model specific to the MEN with full spatial details of the regulatory network was missing. This manuscript addresses this gap and constructs such a model for the MEN.

The authors start out by developing the framework for constructing a compartmental logical model of the MEN. A logical model is chosen over ODE models for its simplicity and scalability for a complex network such as the MEN and its regulatory network. However, this choice constrains the representation of the system to digitalized states and Boolean algebra, and as a result it is difficult to construct/interpret the model for certain protein nodes and mutants. Another assumption that the authors make to construct their model, perhaps for simplicity, is that MEN signaling only occurs at the SPB that enters the bud/daughter cell (dSPB). The asymmetry of the two SPBs in MEN protein localization has been documented extensively and different models have been proposed to address where (which SPB(s) or off the SPBs (the cytosol)) MEN signaling occurs. Nevertheless, both Cdc15 and Dbf2-Mob1 have been shown to localize to the mSPB (the SPB stays in the mother) in addition to dSPB in early anaphase (Molk et al. 2004, Luca et al. 2001, Campbell et al. 2020) while Tem1 only localizes to the mSPB in late anaphase depending on Cdc15. This is in direct contrast to the localization pattern presented in their model (Fig. 5E-F).

The authors then refine the model based on a few key observations/mutant phenotypes. First, they look at CDC15-7A and MOB1-2A which bypass CDK inhibition on Cdc15/Mob1. To explain the SPoC (spindle position checkpoint) defect of CDC15-7A, they updated the regulation of Cdc15's SPB localization to include CDK inhibition where either active Tem1 or lack of CDK inhibition is sufficient for recruitment of Cdc15 to the SPB. This model predicts that Cdc15-7A should be able to bypass a tem1� and localizes to the SPB independently of Tem1. Testing these predictions experimentally would be helpful to validate this unconventional model assumption/setup. They then introduce an anaphase phase specific component (ASC) to fully explain the metaphase behavior of CDC15-7A+MOB1-2A double mutant and later propose that the polo-like kinase Cdc5 is the ASC based on previous experimental observations. Although it is known that SPB localization of Cdc15 depends on Cdc5 in addition to Tem1, this has not been tied to the observation that Cdc5 only localizes to the outer plaque of dSPB after anaphase onset. Thus, the insight that Cdc5 is an anaphase specific regulator of Cdc15 is novel and significant. However, it is misleading to start the model without incorporating or at least mentioning the known fact that Cdc5 is required for the recruitment of Cdc15 to the SPBs.

Next, the authors continue to refine their model with phenotypes of bub2�/bfa1� and kin4�+spo12� double mutant. For bub2/bfa1�, they introduce a new node for Tem1 (Tem1_high) that can simulate the metaphase exit of bub2/bfa1�. However, this model makes a few assumptions that are not well supported by experimental observations. First, in bub2/bfa1�, Tem1 does not accumulate on SPBs to notable levels until late anaphase (Caydasi et al. 2012). It is possible that the low amount of active Tem1 (~75 molecules) at SPBs observed in bub2/bfa1� is sufficient to activate Cdc15 as assumed by Caydasi et al. and the authors here. However, this is not clearly indicated in the text. Second, the evidence for Tem1's direct involvement in recruiting Mob1 to SPBs is weak. Tethering Cdc15 to the SPBs is sufficient to bypass Tem1 (Rock and Amon 2011) and recruit Mob1 (Rock et al. 2013), indicating that Cdc15 alone is responsible for recruiting Mob1 to the SPBs. It is possible that hyperactivated Tem1 in bub2/bfa1� can hyperactivate Cdc15 which then leads to activation of Dbf2-Mob1 in metaphase without invoking Tem1's direct role in recruiting Mob1. In fact, this is consistent with the updated condition used in model 5 for Mob1.SPB (Cdc15_high.SPB & Nud1A & Mob1.Cytoplasm & Tem1_highA). Lastly, CDK inhibition on Mob1 was shown to reduce Dbf2-Mob1's kinase activity (Konig et al. 2010) rather than localization to SPBs as assumed in the model. This is further supported by the ability of Cdc15-SPB to recruit Mob1 to SPBs at any cell cycle stages (Rock et al. 2013). This modification should not affect the model predictions for most mutants but will more closely reflect the experimental observations.

When refining the model with kin4�+spo12� double mutant which restores the SPoC defect from kin4� single mutant, they rediscover that Lte1 plays additional role other than regulating Kin4 as shown by Falk et al. 2016 and Caydasi et al. 2017. Interestingly, they attribute the additional role of Lte1 to the regulation of Bfa1 rather than Tem1 using the phenotype of GAL-KIN4. They simulate two models where Lte1 either inhibits Bfa1's activity or promotes both the activity and SPB localization of Tem1 for GAL-KIN4 (Fig. S2). The latter incorrectly predicts the mitotic exit of anaphase cells with aligned spindle in GAL-KIN4 and is thus ruled out. Have the authors considered a model where Lte1 only promotes Tem1 activity but not localization to SPB? In this model, GAL-KIN4 would in theory inhibit mitotic exit in anaphase cells with aligned spindle given the ability of Kin4 to delocalize Bfa1 and Tem1 from dSPB in this mutant. 

Finally, the refined model was validated with 140 mutant phenotypes and successfully predicted > 80% of the phenotypes. Using this final model, the authors make predictions for forced-localization mutants and experimentally test the effect of forced SPB localization of Tem1, Cdc15 and CDK. All three cases have been tested in literature to some degree with different methods and the results shown here are consistent with previous observations. The model successfully predicted the outcome for SPB tethered Cdc15 and CDK but not for Tem1. It is unclear to us why the model predicts mitotic exit in metaphase for TEM1-SPB since both Tem1's activity (inhibited by Bub2-Bfa1) and its downstream kinases' are restricted by high CDK activity in metaphase in the model. Could the authors provide more information to help the readers understand this more intuitively? 

In addition to the steady state logical model, the authors have also parameterized their model to incorporate continuous timing. This allows them to simulate the effect of FEAR mutants in the timing of mitotic exit and reveals an intersecting insight that has not been explicitly characterized previously regarding the role of the FEAR network in reducing cell-cell variation of mitotic exit. This is further confirmed with experimental measurements of anaphase length for FEAR mutant spo12�. They then go on to apply this continuous model to SPoC mutants and successfully explain the variations observed for SPoC mutants based on the time it takes to exit mitosis relative to spindle alignment. 

In sum, this manuscript provides a valuable framework to synthesize and test observations and models from previous literature as well as future research of the MEN and contributes a step forward for the field. We recommend publication but the authors may want to consider the points raised beforehand.

Additional comments:

1. As mentioned above, the model in Fig. 1 does not include up-to-date information on the MEN, such as regulation of Cdc15 by Cdc5 and an additional Kin4-independent role of Lte1.

2. CDC15-7A +/- MOB1-2A are SPoC defective but not 100%. Could the authors simulate these mutants in their parameterized continuous model used to simulate SPoC mutants in Fig. 7 and compare the results with experimental data (Konig et al. 2010) to validate their model?

3. The role of Cdc5_SPB in model 5 is unclear. In Fig.4 (model 5, which appears to be almost identical as model 3a in Fig. S2), Cdc5_SPB is required for Cdc15 and thus MEN activation. How does this requirement reconcile with the model prediction for CDC15-7A+MOB1-2A double mutant where mitotic exit occurs in metaphase (when spindle migrates into the bud) without Cdc5_SPB which only occurs after anaphase onset (as described in model 3a in Fig. S2)? Adding to the confusion, there are two different rules for Cdc15_high_SPB in model 5 described in Supplementary File 3: 

a. (Nud1.SPB & Tem1_lowA & Cdc15_low.SPB & Cdc5A) | (Nud1.SPB & Tem1_highA & Cdc15_low.SPB) 

b. (Nud1L.SPB & Cdc15_lowL.SPB & ! CDK_lowA.Cytoplasm & Cdc5A.SPB) | (Nud1L.SPB & Cdc15_lowL.SPB & ! CDK_lowA.Cytoplasm & Tem1_lowA.SPB)

, where (a) is for Cdc15_high.SPB in the "Localization" tab and (b) is for Cdc15_highL.SPB in "LocSpecific" tab. In (a), both Tem1_lowA and Cdc5A are required in wild-type cells for Cdc15 to localize to SPBs as described by previous experimental observations (Rock et al. 2011) while in (b) either Cdc5A.SPB or Tem1_lowA.SPB is required which would resolve the issue described above for CDC15-7A+MOB1-2A but contradicts the experimental data. In addition, it appears that inhibition by CDK_low is present in both branches of (b). Which one of the three rules was used in model 5? Based on the results of model 5 for CDC15-7A+MOB1-2A, it seems that rule (b) was likely used. If so, this is not clearly indicated in the figure and is not justified since previous data show clearly that both Cdc5 and Tem1 is required for recruitment of wild-type Cdc15. A potential experiment to resolve the issue raised here with CDC15-7A+MOB1-2A is to test whether Cdc5 is required for Cdc15-7A to localize to SPBs.

4. Similarly, the rules for Lte1 and Cdc5 in regulating Bfa1 activity are inconsistent between Fig. 4 (a, a simplified diagram without locations) and Supplementary File 3 (b):

a. (! Cdc5) | (! Cdc14 & CDK_low & ! Lte1)

b. (! Cdc5A.SPB & ! Lte1A.Cytoplasm & Bfa1_lowA.SPB) | (! Cdc14_lowA.SPB & CDK_lowA.Cytoplasm & Bfa1_lowA.SPB & ! Lte1A.Cytoplasm)

Based on the text in section 4.1.3 and results of model 5, it seems that rule (b) is implemented in model 5 which puts Lte1 in both branches and thus prioritizes Lte1 over Cdc5 and Cdc14. However, this is not clearly stated or represented in the main figure/text and is not formally justified. This model predicts that Bfa1 is inhibited as long as in the same compartment with Lte1 in all cell cycle stages, which is an interesting proposal but remains to be tested. It also appears that this rule (b) serves as the foundation for the statement "In this version of the model the essential role of Cdc5 is only in the recruitment of Cdc15 and not in Bfa1 inhibition". However, just like for Bfa1 inhibition (where Cdc5 is only required in the absence of Lte1 and Cdc14), Cdc5 is no longer essential for the recruitment of Cdc15 in model 5 either (see comment #3, where Cdc5 is only required in the absence of active Tem1). 

5. Localization and activity of Kin4 in metaphase. It appears in the models, Kin4 is assumed to be active and localized to SPBs in the mother cell already in metaphase which does not match the experimental observations (Chan and Amon 2009). 

6. As mentioned earlier, the localization patterns assumed and predicted by the model do not match experimental observations for most of the proteins listed (Fig. 5E-F).

Cdc5: Cdc5's localization to dSPB in late anaphase depends on Bfa1 (Botchkarev et al 2017).

Kin4: does not localized to SPBs in metaphase (Chan and Amon 2009).

Dbf2-Mob1: localizes to both SPBs in anaphase cells with correctly aligned spindles (Luca et al 2001, Campbell et al 2020, Caydasi et al 2012).

Bub2-Bfa1: localization to SPBs is reduced in anaphase cells with mispositioned spindles (Caydasi et al 2009, 2012).

Tem1: in bub2�/bfa1�, Tem1 doesn't accumulate on SPBs to notable levels until late anaphase (after mitotic exit, Caydasi et al. 2012).

Cdc14: localizes mainly to the dSPB in anaphase, likely through Bfa1 (Pereira et al 2002). 

7. The phrase "early/late anaphase" typically does not refer to "pre/post-spindle alignment" since in wild-type cells anaphase onset normally coincides with movement of a SPB into the bud (spindle alignment). Rather, it refers to two different phases of anaphase defined by the spindle length.

Minor comments:

1. It would be helpful for the readers to include Kin4's role in regulating Bfa1's SPB localization in Figure 4.

2. What is the basis for assuming Cdc15-phosphorylated Nud1 is the receptor for CDK at SPBs? In addition, what is the basis for the assumption that MEN activity (here assumed to be active Dbf2-Mob1) excludes CDK from SPB? At mSPB where Dbf2-Mob1 and CDK are both present in anaphase, this is clearly not the case.

0. Line 451&452 should be Fig. 5E and F.

1. The Supplementary Files 3 and 5 could use some descriptions for the labels (abbreviations) to help readers understand easier. 

2. The figure # in Supplementary Files 5 is shifted by one (i.e., Fig. 2 is actually Fig. 3).

Reviewer #3 (Adrien Fauré, signed review): This is a review of the manuscript "Unifying the mechanism of mitotic exit control in a spatio-temporal logical model" by Howell and colleagues. The paper introduces a logical model of the mitotic exit network (MEN), incorporating a wealth of detail, particularly regarding spatial localization of the various proteins to the nucleus, cytoplasm, bud as well as the mother- and daughter-bound spindle pole bodies. This model alone represents a valuable contribution to the field, but the authors did not stop there and could actually check some of the models predictions, showing that Cdc5 overexpression can not rescue mob1∆ mutants, that SPO12 has a role in controlling the robustness of anaphase length, or testing the recruitment of CDC15, Tem1 or Cdc28 to the SPB, thus bringing new experimental results and valuable insight into the biology of mitotic exit in the budding yeast. Moreover, some of those experiments lead to interesting methodological discussions about the strengths and limitations of the logical framework, particularly regarding overexpression and localisation mutants and discretization. On all three aspects — review of the literature and modeling, experimental predictions and tests, methodological development, I found the paper extremely compelling.

In addition I should mention that the paper is very well written. The relevant literature on the MEN is clearly presented, the logical framework and previous models discussed in (almost?) all relevant details, model construction and refinement described step by step (with one possible oversight, see below). Figures are clear. A very pleasant read all in all.

On the minus side, there's not much to say. My main regret is the model itself was not added as supplementary material — unless I missed it? Everything else seems to be there, including R and python scripts, there's a table with the logical rules, and all the relevant data I think, but the reader who would like to play with the model would have to copy everything in the relevant format, a tedious task that stopped me from actually checking the model myself during this review. I trust the model files will be included in the final version, alongside with model annotations etc., possibly in the BoolNet or GINsim formats used by the authors, or the SBML-qual format.

In the introduction, the authors mention the tool EpiLog and then write, "there is currently no logical formalism capable of representing intracellular spatial regulation." This is I think a bit of a stretch, as indeed, although EpiLog itself only deals with multicellular systems, it relies on previous, more general work on model composition (see Mendes et al., Bioinformatics. 2013; 29(6): 749-57). To the best my knowledge this has only been applied to multicellular systems, but it could apply just as well to intracellular compartment, and I believe it deserves to be discussed.

On a similar note — there are different, albeit related, approaches to represent a logical model as a Boolean network (section 3.5). I don't think the chosen approach should influence the results, but I would still be curious to know how the authors proceeded.

Finally, there's also a few minor points, missing (or sometimes extra) commas, words (l. 167, 768) or apostrophes (l. 144) here and there. Longer sentences, especially those with "however", can be a little hard to parse and could be rephrased. Some abbreviations are introduced before or without their definition (dSPB in section 3.3, the "GEF" later). The lines that refer to Fig. 8C and S6C appear redundant and it seems the two figures could be mentioned together. And unless I'm mistaken, "If there are n proteins represented in the model and C compartments, the resulting compartmental logical network" should have n × C nodes, and not 2 × n × C (I mean, if there are n protein and only one compartment, there would be only n nodes, right?)

And I think that's about all? I'm happy to recommend this paper for publication after minor revision.

Best regards,

---

## [Decision Letter · Decision Letter 2]

29 Aug 2020

Dear Dr Csikász-Nagy,

Thank you for submitting your revised Research Article entitled "Unifying the mechanism of mitotic exit control in a spatio-temporal logical model" for publication in PLOS Biology. I have now obtained advice from the three original reviewers and have discussed their comments with the Academic Editor. 

Based on the reviews, we will probably accept this manuscript for publication, assuming that you will modify the manuscript to address the remaining point raised by Reviewer 3 (see comments below). Please also make sure to address the data and other policy-related requests noted at the end of this email.

We expect to receive your revised manuscript within two weeks. Your revisions should address the specific point made by Reviewer 3. In addition to the remaining revisions and before we will be able to formally accept your manuscript and consider it "in press", we also need to ensure that your article conforms to our guidelines. A member of our team will be in touch shortly with a set of requests. As we can't proceed until these requirements are met, your swift response will help prevent delays to publication.

*Copyediting*

*Published Peer Review History*

*Early Version*

*Submitting Your Revision*

Sincerely,

Ines

--

Ines Alvarez-Garcia, PhD,

Senior Editor,

ialvarez-garcia@plos.org,

PLOS Biology

Fig. 6A, D, E; Fig. 7A, C, D, E, F; Fig. S5A, B, C; Fig. S6A, B, C, D and Fig. S8A, D

In File S7 you mention that the data shown in Fig. 6 is included, but we are unsure if the data underlying the graphs shown in Fig. 6A, D and E has been included. If so, please label clearly the data, otherwise please provide it.

Reviewers’ comments

Rev. 1:

The authors addressed all the concerns and suggestions and it is suitable for publication.

Rev. 2: Angelika Amon and Xiaoxue Snow Zhou

The authors have addressed our comments

Rev. 3: Adrien Fauré

The authors have answered most of my comments — except one point, where I should have been clearer:

> This is described in the methods section: "We represented the logical model as a Boolean network, in which each level of activity is represented as an individual node, in order to make use of computational tools designed for Boolean networks."

There are many possible variations on the "each level of activity is represented as an individual node" method.

GINsim / bioLQM has one implementation: http://doc.ginsim.org/lrg-modifier-bool.html
http://colomoto.org/biolqm/doc/modifier-booleanization.html

but many others are possible (see Didier et al., Mapping multivalued onto Boolean dynamics, JTB 270:177-184, 2011; and Tonello, On the conversion of multivalued to Boolean dynamics, Discrete Applied Mathematics 259:193-204, 2019)

Importantly, some of the variants considered by Didier et al. do not guaranty that non-admissible states are not reachable from admissible initial states, contrary to bioLQM's implementation, and may generate non-admissible attractors.

---

## [Editor Report · Decision Letter 3]

9 Oct 2020

Dear Dr Csikász-Nagy,

On behalf of my colleagues and the Academic Editor, Jonathon Pines, I am pleased to inform you that we will be delighted to publish your Research Article in PLOS Biology. 

Early Version

PRESS 

Kind regards,

Erin O'Loughlin

Publishing Editor, 

PLOS Biology

on behalf of

Ines Alvarez-Garcia,

Senior Editor

PLOS Biology